# WHY CLEAN GENERALIZATION AND ROBUST OVERFITTING BOTH HAPPEN IN ADVERSARIAL TRAINING

## ABSTRACT

Adversarial training is a standard method to train deep neural networks to be robust to adversarial perturbation. Similar to surprising *clean generalization* ability in the standard deep learning setting, neural networks trained by adversarial training also generalize well for *unseen clean data*. However, in constrast with clean generalization, while adversarial training method is able to achieve low robust training error, there still exists a significant *robust generalization gap*, which promotes us exploring what mechanism leads to both *clean generalization and robust overfitting (CGRO)* during learning process. In this paper, we provide a theoretical understanding of this puzzling phenomenon (CGRO) through *feature learning theory*. Specifically, we prove that, under our theoretical framework (patch-structured dataset and one-hidden-layer CNN model) , a *three-stage phase transition* happens from adversarial training dynamics, and the network learner provably partially learns the true feature but exactly memorizes the spurious features from training-adversarial examples, which thereby results in CGRO phenomenon. Besides, for more general data assumption, we then show the efficiency of CGRO classifier from the perspective of *representation complexity*. On the empirical side, we also verify our theoretical analysis about learning process in real-world vision dataset.

## 1 INTRODUCTION

Nowadays, deep neural networks have achieved excellent performance in a variety of disciplines, especially including in computer vision (Krizhevsky et al., 2012; Dosovitskiy et al., 2020; Kirillov et al., 2023) and natural language process (Devlin et al., 2018; Brown et al., 2020; Ouyang et al., 2022). However, it is well-known that small but adversarial perturbations to the natural data can make well-trained classifiers confused (Biggio et al., 2013; Szegedy et al., 2013; Goodfellow et al., 2014), which potentially gives rise to reliability and security problems in real-world applications and promotes designing adversarial robust learning algorithms.

In practice, adversarial training methods (Goodfellow et al., 2014; Madry et al., 2017; Shafahi et al., 2019; Zhang et al., 2019; Pang et al., 2022) are widely used to improve the robustness of models by regarding perturbed data as training data. However, while these robust learning algorithms are able to achieve high robust training accuracy (Gao et al., 2019), it still leads to a non-negligible robust generalization gap (Raghunathan et al., 2019), which is also called *robust overfitting* (Rice et al., 2020; Yu et al., 2022).

To explain this puzzling phenomenon, a series of works have attempted to provide theoretical understandings from different perspectives. Despite these aforementioned works seem to provide a series of convincing evidence from theoretical views in different settings, there still exists *a gap between theory and practice* for at least *two reasons*.

*First*, although previous works have shown that adversarial robust generalization requires more data and larger models (Schmidt et al., 2018; Gowal et al., 2021; Li et al., 2022; Bubeck & Sellke, 2023), it is unclear that what mechanism, during adversarial training process, *directly* causes robust overfitting. A line of work about uniform algorithmic stability (Xing et al., 2021; Xiao et al., 2022), under Lipschitzian smoothness assumptions, also suggest that robust generalization gap increases when training iteration is large. In other words, we know there is no robust generalization gap for a trivial model that only guesses labels totally randomly (e.g. deep neural networks at random initialization), which implies that we should take learning process into consideration to analyze robust generalization.

*Second and most importantly*, while some works (Tsipras et al., 2018; Zhang et al., 2019; Hassani & Javanmard, 2022) point out that achieving robustness may hurt clean test accuracy, in most of the cases, it is observed that drop of robust test accuracy is much higher than drop of clean test accuracy in adversarial training (Madry et al., 2017; Schmidt et al., 2018; Raghunathan et al., 2019) (see in Figure 1, where clean test accuracy is more than $80\%$ but robust test accuracy only attains nearly $50\%$). Namely, a weak version of benign overfitting (Zhang et al., 2017), which means that overparameterized deep neural networks can both fit random data powerfully and generalize well for unseen *clean* data, remains after adversarial training.

Therefore, it is natural to ask the following question:

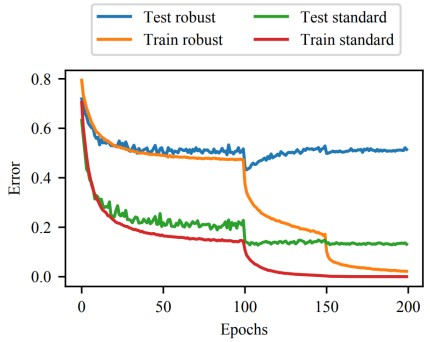

Figure 1: The learning curves of adversarial training on CIFAR 10 (Rice et al., 2020) .

*What is the underlying mechanism that results in both Clean Generalization and Robust Overfitting (CGRO) during adversarial training?*

In this paper, we provide a theoretical understanding of this question. We propose a theoretical framework of adversarial training under which we analyze feature learning process to explain why deep neural networks trained by adversarial training have a good clean test performance but a poor robust generalization at the same time.

First, we present the *existence* of *CGRO classifiers* that achieve *both clean generalization and robust overfitting* in a general setting. Specifically, we assume that there exists a *clean classifier* $f_{\text{clean}}$ that can perfectly classify the natural data but fails to classify the adversarially perturbed data, which is consistent with the common practice that well-trained neural networks are vulnerable to adversarial examples (Szegedy et al., 2013; Raghunathan et al., 2019). Besides, we also assume that the supporting set of data distribution $\mathcal{D}$ is *well-separated*, which means half of the distance between the positive data and negative data is larger than the $\ell_p$ perturbation radius $\delta$ (Yang et al., 2020; Li et al., 2022). The training dataset $\mathcal{S} = \{(\boldsymbol{X}_1, y_1), (\boldsymbol{X}_2, y_2), \ldots, (\boldsymbol{X}_N, y_N)\}$ with $N-$samples are randomly drawn from the data distribution $\mathcal{D}$. Then, we consider the following function as

$$f_{\mathcal{S}}(\boldsymbol{X}) = \underbrace{f_{\text{clean}}(\boldsymbol{X}) \left(1 - \mathbb{I}\{\boldsymbol{X} \in \cup_{i=1}^{N} \mathbb{B}_p(\boldsymbol{X}_i, \delta)\}\right)}_{\text{clean classification on unseen test data}} + \underbrace{\sum_{i=1}^{N} y_i \mathbb{I}\{\boldsymbol{X} \in \mathbb{B}_p(\boldsymbol{X}_i, \delta)\}}_{\text{robust classification on training data}}. \quad \text{(C)}$$

where $\mathbb{B}_p(\mathbf{z}, r)$ denotes the neighborhood of the center $\mathbf{z}$ with the $\ell_p-$radius $r$ and $\mathbb{I}\{\mathcal{A}\}$ denotes the standard indicator of the event $\mathcal{A}$.

Indeed, it is clear that $f_{\mathcal{S}}$ belongs to CGRO classifier, which perfectly clean classifies unseen test data by the first summand and robustly memorizes training data by the second summand.

Theorefore, inspired by the intuition of the $f_{\mathcal{S}}$'s construction, we conjecture that neural networks trained by adversarial training converge to the similarities of $f_{\mathcal{S}}$ that can be decomposed into two components, which correspond to clean generalization and robust overfitting respectively.

To further demonstrate our conjecture, we theoretically and empirically investigate the learning process of adversarial training in this work. More precisely, we make the following contributions:

- In Section 3, we introduce a theoretical framework of adversarial training, where we leverage patch-structured dataset as a simplification of real-world image data and choose non-linear one-hidden-layer CNN model as the network learner .

- In Section 4, under our theoretical framework, we apply a *signal-noise decomposition* to characterize the feature learning process. We then propose a novel *three-stage* analysis technique to decouple the complicated training dynamics as follows.

  - **Stage I:** First of all, the signal component increases quadratically at initialization, and the model starts to learn *partial true feature*.

- **Stage II:** Afterwards, the growth of signal component *nearly stops updating* since that the increment of signal component is now mostly dominated by the noise component.

- **Stage III:** Eventually, by the quadratic increment of noise component, the model *exactly memorizes* the *data-wise* spurious feature.

As a consequence, it leads the network learner to CGRO regime.

- In Section 5, with more general data assumptions, we study CGRO classifiers via the view of representation complexity. We prove that achieving CGRO classifier only needs *polynomial* complexity but robust classifier requires even *exponential* complexity in worst case.

- In Section 6, in order to verify our conjecture in real-world dataset, we empirically investigate the dynamics of loss landscape over input during learning process on real-image datasets MNIST and CIFAR10. The experiment results suggest that the model trained by adversarial training tends to memorize data by approximating local robust indicators on training data.

**Notations.** In this work, we use lower case letters for scalars, lower case bold for vectors. We use $\mathrm{poly}(\cdot)$ to denote the polynomial order, and $\mathrm{polylog}(\cdot)$ to denote the some polynomial in log order. We use $O(\cdot), \Omega(\cdot)$ to hide absolute constants which do not depend on any problem parameter, and $\tilde{O}(\cdot)$ to hide absolute constants and log factors. For a distribution $\mathcal{D}$, the set of adversarial examples $\mathcal{S}^{\mathrm{adv}}$, $\ell_p-$perturbation radius $\delta$ and classifier $f$, we define the clean test error as $\mathbb{E}_{(\boldsymbol{X},y)\sim\mathcal{D}}[\mathbb{I}\{yf(\boldsymbol{X}) \leq 0\}]$, the robust test error as $\mathbb{E}_{(\boldsymbol{X},y)\sim\mathcal{D}}[\max_{\|\boldsymbol{\xi}\|_p\leq\delta} \mathbb{I}\{yf(\boldsymbol{X}+\boldsymbol{\xi}) \leq 0\}]$ and the robust training error as $\frac{1}{|\mathcal{S}^{\mathrm{adv}}|}\sum_{(\boldsymbol{X}^{\mathrm{adv}},y)\in\mathcal{S}^{\mathrm{adv}}} \mathbb{I}\{yf(\boldsymbol{X}^{\mathrm{adv}}) \leq 0\}$. ReLU activation function is defined as $\max(0,\cdot)$.

## 2 ADDITIONAL RELATED WORK

**Empirical Works on Robust Overfitting.** One surprising behavior of deep learning is that over-parameterized neural networks can generalize well, which is also called *benign overfitting* that deep models have not only the powerful memorization but a good performance for unseen data (Zhang et al., 2017; Belkin et al., 2019). However, in contrast to the standard (non-robust) generalization, for the robust setting, Rice et al. (2020) empirically investigates robust performance of models based on adversarial training methods, which are used to improve adversarial robustness (Szegedy et al., 2013; Madry et al., 2017), and the work shows that *robust overfitting* can be observed on multiple datasets.

**Theoretical Works on Robust Overfitting.** A list of works (Schmidt et al., 2018; Balaji et al., 2019; Dan et al., 2020) study the *sample complexity* for adversarial robustness, and their works manifest that adversarial robust generalization requires more data than the standard setting, which gives an explanation of the robust generalization gap from the perspective of statistical learning theory. And another line of works (Tsipras et al., 2018; Zhang et al., 2019) propose a principle called the *robustness-accuracy trade-off* and have theoretically proven the principle in different setting, which mainly explains the widely observed drop of robust test accuracy due to the trade-off between adversarial robustness and clean accuracy. Recently, Li et al. (2022) investigates the robust expressive ability of neural networks and shows that robust generalization requires exponentially large models.

**Feature Learning Theory of Deep Learning.** The *feature learning theory* of neural networks (Allen-Zhu & Li, 2020a;b; 2022; Shen et al., 2022; Jelassi & Li, 2022; Jelassi et al., 2022; Chen et al., 2022) is proposed to study how features are learned in deep learning tasks, which provide a theoretical analysis paradigm beyond the *neural tangent kernel (NTK) theory* (Jacot et al., 2018; Du et al., 2018; 2019; Allen-Zhu et al., 2019; Arora et al., 2019). In this work, we make a first step to understand clean generalization and robust overfitting (CGRO) phenomenon in adversarial training by analyzing feature learning process under our theoretical framework.

## 3 A THEORETICAL FRAMEWORK OF ADVERSARIAL TRAINING

In this section, we introduce a theoretical framework of adversarial training under which we can directly analyze the learning process to provide a theoretical explanation to CGRO phenomenon.

## 3.1 STRUCTURED DATASET

The data's patch structure that we leverage can be viewed as a simplification of real-world vision-recognition datasets (Chen et al., 2021; Jelassi & Li, 2022; Jelassi et al., 2022; Kou et al., 2023). Specifically, we consider the binary classification data with the following patch structure as

**Patch Data Distribution.** We define a data distribution $\mathcal{D}$, in which each instance consists in an input $\boldsymbol{X} \in \mathbb{R}^D$ and a label $y \in \{-1, 1\}$ generated by

1. The label $y$ is uniformly drawn from $\{-1, 1\}$.
2. The input $\boldsymbol{X} = (\boldsymbol{X}[1], \dots, \boldsymbol{X}[P])$, where each patch $\boldsymbol{X}[j] \in \mathbb{R}^d$ and $P = D/d$ is the number of patches (we assume that $D/d$ is an integer and $P = \text{polylog}(d)$).
3. Meaningful Signal patch: for each instance, there exists one and only one meaningful patch $\text{signal}(\boldsymbol{X}) \in [P]$ satisfies $\boldsymbol{X}[\text{signal}(\boldsymbol{X})] = \alpha y \boldsymbol{w}^*$, where $\alpha \in \mathbb{R}_+$ is the norm of data and $\boldsymbol{w}^* \in \mathbb{R}^d (\|\boldsymbol{w}^*\|_2 = 1)$ is the unit meaningful signal vector.
4. Noisy patches: $\boldsymbol{X}[j] \sim \mathcal{N}\left(0, \left(\mathbf{I}_d - \boldsymbol{w}^*\boldsymbol{w}^{*\top}\right)\sigma^2\right)$, for $j \in [P] \backslash \{\text{signal}(\boldsymbol{X})\}$.

We assume $\alpha = d^{0.249} \text{polylog}(d), \sigma = d^{-0.509}$ to enable meaningful signal is stronger than noise. Indeed, images are divided into signal patches that are meaningful for the classification such as the whisker of a cat or the nose of a dog, and noisy patches like the uninformative background of a photo.

## 3.2 LEARNER NETWORK

To learn our synthetic dataset, we use a one-hidden layer convolutional neural network (CNN) (LeCun et al., 1998; Krizhevsky et al., 2012) with non-linear activation as the learner network.

**Simplified CNN Model.** For a given input data $\boldsymbol{X}$, the model outputs as

$$f_{\boldsymbol{W}}(\boldsymbol{X}) = \sum_{r=1}^{m} \sum_{j=1}^{P} \sigma\left(\langle \boldsymbol{w}_r, \boldsymbol{X}[j] \rangle\right). \tag{M}$$

The first layer weights are $\boldsymbol{W} \in \mathbb{R}^{m \times d}$ and the second layer is fixed to $\mathbf{1}_m$. And we apply the cubic activation function $\sigma(z) = z^3$, as polynomial activations are standard in literatures of deep learning theory (Kileel et al., 2019; Allen-Zhu & Li, 2020a;b; Jelassi & Li, 2022; Jelassi et al., 2022).

**The Role of Non-linearity.** Indeed, a series of recent theoretical works (Li et al., 2019; Chen et al., 2021; Javanmard & Soltanolkotabi, 2022) show that linear model can achieve robust generalization for adversarial training under certain settings, which but fails to explain the CGRO phenomenon observed in practice. To mitigate this gap, we improve the expressive power of model by using non-linear activation that can characterize the data structure and learning process more precisely.

## 3.3 ADVERSARIAL TRAINING

In adversarial training, with access to the training dataset $\mathcal{S} = \{(\boldsymbol{X}_1, y_1), (\boldsymbol{X}_2, y_2), \dots, (\boldsymbol{X}_N, y_N)\}$ (where we assume $N = \text{poly}(d)$) randomly sampled from the data distribution $\mathcal{D}$, we aim to minimize the following adversarial loss that is a trade-off between natural risk and robust regularization.

**Adversarial Loss.** For a hyperparameter $\lambda > 0$, the adversarial loss is defined as

$$\frac{1}{N} \sum_{i=1}^{N} \underbrace{\mathcal{L}\left(\boldsymbol{W}; \boldsymbol{X}_i, y_i\right)}_{\text{natural risk}} + \lambda \cdot \underbrace{\max_{\widehat{\boldsymbol{X}}_i \in \mathbb{B}_p(\boldsymbol{X}_i, \delta)} \left[\mathcal{L}\left(\boldsymbol{W}; \widehat{\boldsymbol{X}}_i, y_i\right) - \mathcal{L}\left(\boldsymbol{W}; \boldsymbol{X}_i, y_i\right)\right]}_{\text{robust regularization}}. \tag{F}$$

where we use $\mathcal{L}(\boldsymbol{W}; \boldsymbol{X}, y)$ to denote the single-point loss with respect to $f_{\boldsymbol{W}}$ on $(\boldsymbol{X}, y)$, and assume the $\ell_p-$perturbation radius $\delta = \alpha \left(1 - \frac{1}{\sqrt{d}\, \text{polylog}(d)}\right)$.

This adversarial loss gives a general form of adversarial training methods (Goodfellow et al., 2014; Madry et al., 2017; Zhang et al., 2019) for different values of hyperparameter $\lambda$, where we assume $\lambda \in [\frac{1}{\text{poly}(d)}, 1)$. Then, by using the logistic loss, we derive the following training objective.

**Training Objective.** For a hyperparameter $\lambda > 0$, the training objective is defined as

$$\widehat{\mathcal{L}}_{\mathrm{adv}}(\boldsymbol{W}) = \frac{1}{N}\sum_{i=1}^{N}(1-\lambda)\log\left(1 + e^{-y_i f_W(\boldsymbol{X}_i)}\right) + \lambda\log\left(1 + e^{-y_i f_W\left(\boldsymbol{X}_i^{\mathrm{adv}}\right)}\right). \qquad \text{(O)}$$

where we apply an adversarial attack method $\mathrm{Attack}(\boldsymbol{X}, y; p, \delta)$ to generate adversarial examples

$$\boldsymbol{X}_i^{\mathrm{adv}} = \mathrm{Attack}(\boldsymbol{X}_i, y_i; p, \delta) \in \mathbb{B}_p(\boldsymbol{X}_i, \delta),$$

for $i \in [N]$, in order to approximate $\mathrm{argmax}_{\widehat{\boldsymbol{X}}_i \in \mathbb{B}_p(\boldsymbol{X}_i, \delta)}\mathcal{L}(\boldsymbol{W}; \widehat{\boldsymbol{X}}_i, y_i)$ in (F). To simplify our analysis of learning process, we consider the following $\ell_2-$adversarial attack method.

**Geometry-inspired Transferable Attack (GTA).** For a given instance $(\boldsymbol{X}, y)$ and a target classifier $g$, the algorithm outputs an adversarial example as

$$\boldsymbol{X}^{\mathrm{adv}} = \boldsymbol{X} - \gamma\frac{g(\boldsymbol{X})}{\|\nabla_{\boldsymbol{X}}g(\boldsymbol{X})\|_2}\nabla_{\boldsymbol{X}}g(\boldsymbol{X}), \qquad \text{(A)}$$

where $\gamma > 0$ is a scalar to enable $\boldsymbol{X}^{\mathrm{adv}} \in \mathbb{B}_2(\boldsymbol{X}, \delta)$.

Geometry-inspired adversarial attack is a computing-efficient and loss-free attack method (Moosavi-Dezfooli et al., 2016; Tursynbek et al., 2022), which has a comparable adversarial performance with FGSM (Goodfellow et al., 2014) and PGD (Kurakin et al., 2018). And we also leverage the transferability of adversarial examples (Papernot et al., 2016; Charles et al., 2019; Ilyas et al., 2019)by designing a target classifier $g$, which helps us focus on feature learning process of our CNN model.

**Training Algorithm.** To solve the minimization problem (O), we run gradient descent (GD) algorithm to update our CNN weights $\boldsymbol{W}$ for $T$ iterations, which is defined as

$$\boldsymbol{W}^{(t+1)} = \boldsymbol{W}^{(t)} - \eta\nabla_{\boldsymbol{W}}\widehat{\mathcal{L}}_{\mathrm{adv}}\left(\boldsymbol{W}^{(t)}\right), \qquad \text{(T)}$$

where $\eta > 0$ is the learning rate.

Next, we present the detailed parameterization setting in adversarial training.

**Parameterization 3.1.** *For our CNN model (M), we set the width $m = \mathrm{polylog}(d)$ to ensure the network is mildly over-parameterized. At initialization, we choose the weights $\boldsymbol{w}_1, \boldsymbol{w}_2, \ldots, \boldsymbol{w}_m$ are i.i.d sampled from the same Gaussian distribution $\mathcal{N}(0, \sigma_0^2\mathbf{I}_d)$, where $\sigma_0^2 = \frac{\mathrm{polylog}(d)}{d}$. For learning process, we set the target classifier $g(\boldsymbol{X}) = \langle\boldsymbol{w}^*, \boldsymbol{X}[\mathrm{signal}(\boldsymbol{X})]\rangle$, the scalar $\gamma = 1 - \frac{1}{\sqrt{d}\,\mathrm{polylog}(d)}$ in (A) and the learning rate $\eta = O(1)$ in (T).*

Under Parameterization 3.1, we choose a linear model $g(\boldsymbol{X}) = \langle\boldsymbol{w}^*, \boldsymbol{X}[\mathrm{signal}(\boldsymbol{X})]\rangle$ as the target classifier for GTA (A), which indeed implies a reasonable attack. Intuitively, when the model $f_{\boldsymbol{W}}$ has achieve mid-high clean test accuracy, the decision boundary of $f_{\boldsymbol{W}}$ will have a significant correlation with the separating plane of $g$, which thus makes adversarial examples generated by $g$ transferable.

## 4 FEATURE LEARNING PROCESS UNDER OUR THEORETICAL FRAMEWORK

In this section, we analyze the feature learning process to understand CGRO in adversarial training.

### 4.1 FEATURE LEARNING

First, we provide an introduction to feature learning, which is widely applied in theoretical works (Allen-Zhu & Li, 2020a;b; 2022; Shen et al., 2022; Jelassi & Li, 2022; Jelassi et al., 2022; Chen et al., 2022) explore what and how neural networks learn in different tasks. In this work, we first leverage feature learning theory to explain CGRO phenomenon in adversarial training. Specifically, for an arbitrary clean training data-point $(\boldsymbol{X}, y) \sim \mathcal{D}$ and a given model $f_{\boldsymbol{W}}$, we focus on

- **True Feature Learning.** We project the weight $\boldsymbol{W}$ on the meaningful signal vector to measure the correlation between the model and the true feature as

$$\mathcal{U} := \sum_{r=1}^{m}\langle\boldsymbol{w}_r, \boldsymbol{w}^*\rangle^3.$$

- **Spurious Feature Learning.** We project the weight $\boldsymbol{W}$ on the random noise to measure the correlation between the model and the spurious feature as

$$\mathcal{V} := y \sum_{r=1}^{m} \sum_{j \in [P] \backslash \mathrm{signal}(\boldsymbol{X})} \langle \boldsymbol{w}_r, \boldsymbol{X}[j] \rangle^3 .$$

We then calculate the model's classification correctness on certain clean data point as

$$
\begin{aligned}
y f_{\boldsymbol{W}}(\boldsymbol{X}) &= y \sum_{r=1}^{m} \langle \boldsymbol{w}_r, \boldsymbol{X}[\mathrm{signal}(\boldsymbol{X})] \rangle^3 + y \sum_{r=1}^{m} \sum_{j \in [P] \backslash \mathrm{signal}(\boldsymbol{X})} \langle \boldsymbol{w}_r, \boldsymbol{X}[j] \rangle^3 \\
&= \underbrace{y \sum_{r=1}^{m} \langle \boldsymbol{w}_r, \alpha y \boldsymbol{w}^* \rangle^3}_{\alpha^3 \mathcal{U}} + \underbrace{y \sum_{r=1}^{m} \sum_{j \in [P] \backslash \mathrm{signal}(\boldsymbol{X})} \langle \boldsymbol{w}_r, \boldsymbol{X}[j] \rangle^3}_{\mathcal{V}} .
\end{aligned}
$$

Thus, the model correctly classify the data if and only if $\alpha^3 \mathcal{U} + \mathcal{V} \geq 0$, which holds in at least two cases. Indeed, one is that the model learns the true feature and ignores the spurious features, where $\mathcal{U} = \Omega(1) \gg |\mathcal{V}|$. Another is that the model doesn't learn the true feature but memorizes the spurious features, where $|\mathcal{U}| = o(1)$ and $|\mathcal{V}| = \Omega(1) \gg 0$.

Therefore, this analysis tells us that the model will generalize well for unseen data if the model learns true feature. But the model will overfit training data if the model only memorizes spurious features since the data-specific random noises are independent for distinct instances, which means that, with high probability, it holds that $\mathcal{V} = o(1)$ for unseen data $(\boldsymbol{X}, y)$.

We also calculate the model's classification correctness on perturbed data point, where we use geometry-inspired transferable attack proposed in (A) to generate adversarial example as

$$\boldsymbol{X}^{\mathrm{adv}}[j] = \boldsymbol{X}[j] - \gamma \frac{g(\boldsymbol{X})}{\|\nabla_{\boldsymbol{X}} g(\boldsymbol{X})\|_2} \nabla_{\boldsymbol{X}} g(\boldsymbol{X})[j] \quad = \left\{ \begin{array}{l} \alpha(1-\gamma) y \boldsymbol{w}^*, j = \mathrm{signal}(\boldsymbol{X}) \\ \boldsymbol{X}[j], j \in [p] \backslash \mathrm{signal}(\boldsymbol{X}) \end{array} \right.$$

We then derive the correctness as

$$y f_{\boldsymbol{W}}(\boldsymbol{X}^{\mathrm{adv}}) = \underbrace{y \sum_{r=1}^{m} \langle \boldsymbol{w}_r, \alpha(1-\gamma) y \boldsymbol{w}^* \rangle^3}_{\alpha^3(1-\gamma)^3 \mathcal{U}} + \underbrace{y \sum_{r=1}^{m} \sum_{j \in [P] \backslash \mathrm{signal}(\boldsymbol{X})} \langle \boldsymbol{w}_r, \boldsymbol{X}[j] \rangle^3}_{\mathcal{V}} .$$

Thus, the model correctly classify the perturbed data if and only if $\alpha^3(1-\gamma)^3 \mathcal{U} + \mathcal{V} \geq 0$, which implies that we can analyze the perturbed data similarly.

## 4.2 MAIN RESULT

Now, we present our main result about feature learning process as the following theorem.

**Theorem 4.1.** *Under Parameterization 3.1, we run the adversarial training algorithm to update the weight of the simplified CNN model for $T = \Omega(\mathrm{poly}(d))$ iterations. Then, with high probability, it holds that the CNN model*

1. *partially learns the true feature, i.e. $\mathcal{U}^{(T)} = \Theta(\alpha^{-3})$;*

2. *exactly memorizes the spurious feature, i.e. for each $i \in [N]$, $\mathcal{V}_i^{(T)} = \Theta(1)$,*

*where $\mathcal{U}^{(t)}$ and $\mathcal{V}_i^{(t)}$ is defined for $i-$th instance $(\boldsymbol{X}_i, y_i)$ and $t-$th iteration as the same in (1)(1). Consequently, the clean test error and robust training error are both smaller than $o(1)$, but the robust test error is at least $\frac{1}{2} - o(1)$.*

Theorem 4.1 states that, during adversarial training, the neural network partially learns the true feature of objective classes and exactly memorizes the spurious features depending on specific training data, which causes that the network learner is able to correctly classify clean data by partial meaningful signal (*clean generalization*), but fails to classify the unseen perturbed data since it leverages only data-wise random noise to memorize training adversarial examples (*robust overfitting*).

## 4.3 ANALYSIS OF LEARNING PROCESS

Next, we provide a proof sketch of Theorem 4.1. To obtain a detailed analysis of learning process, we consider the following objects that can be viewed as weight-wise version of $\mathcal{U}^{(t)}$ and $\mathcal{V}_i^{(t)}$. For $r \in [m]$, $i \in [N]$ and $j \in [P] \setminus \text{signal}(\boldsymbol{X}_i)$, we define $u_r^{(t)}$ and $v_{i,j,r}^{(t)}$ as

**Signal Component.** $u_r^{(t)} := \left\langle \boldsymbol{w}_r^{(t)}, \boldsymbol{w}^* \right\rangle$, thus $\mathcal{U}^{(t)} = \sum_{r \in [m]} \left( u_r^{(t)} \right)^3$.

**Noise Component.** $v_{i,j,r}^{(t)} := y_i \left\langle \boldsymbol{w}_r^{(t)}, \boldsymbol{X}_i[j] \right\rangle$, thus $\mathcal{V}_i^{(t)} = \sum_{r \in [m]} \sum_{j \in [P] \setminus \text{signal}(\boldsymbol{X}_i)} \left( v_{i,j,r}^{(t)} \right)^3$.

**Phase I: At the beginning, the signal component of lottery tickets winner $\max_{r \in [m]} u_r^{(t)}$ increases quadratically (Lemma 4.2). At this point, the model starts to learn partial true feature.**

**Lemma 4.2.** *(Lower Bound of Signal Component Growth) For each $r \in [m]$ and any $t \geq 0$, the signal component grows as*

$$u_r^{(t+1)} \geq u_r^{(t)} + \Theta(\eta \alpha^3) \left( u_r^{(t)} \right)^2 \psi \left( \alpha^3 \sum_{k \in [m]} \left( u_k^{(t)} \right)^3 \right),$$

*where we use $\psi(\cdot)$ to denote the negative sigmoid function $\psi(z) = \frac{1}{1+e^z}$ as well as Lemma 4.3,4.4.*

Lemma 4.2 manifests that the signal component increases quadratically at initialization. Therefore, we know that, after $T_0 = \Theta \left( \frac{1}{\eta \alpha^3 \sigma_0} \right)$ iterations, the maximum signal component $\max_{r \in [m]} u_r^{(T_0)}$ attains the order $\tilde{\Omega}(\alpha^{-1})$, which implies the model learns partial true feature.

**Phase II: Once the maximum signal component $\max_{r \in [m]} u_r^{(t)}$ attains the order $\tilde{\Omega}(\alpha^{-1})$, the growth of signal component nearly stops updating since that the increment of signal component is now mostly dominated by the noise component (Lemma 4.3).**

Due to the property of the negative sigmoid function $\phi(z) = \frac{1}{1+e^z}$, the growth of signal component becomes very slow when $\psi$'s input attains the order $\Omega(1)$. This intuition can be formally represented as follow.

**Lemma 4.3.** *(Upper Bound of Signal Component Growth) For $T_0 = \Theta \left( \frac{1}{\eta \alpha^3 \sigma_0} \right)$ and any $t \in [T_0, T]$, the signal component is upper bounded as*

$$\max_{r \in [m]} u_r^{(t)} \leq \tilde{O}(\alpha^{-1}) + \tilde{O} \left( \frac{\eta \alpha^3 (1-\gamma)^3}{N} \right) \sum_{s=T_0}^{t-1} \sum_{i=1}^{N} \psi \left( \alpha^3 (1-\gamma)^3 \sum_{k \in [m]} \left( u_k^{(s)} \right)^3 + \mathcal{V}_i^{(s)} \right).$$

Lemma 4.3 shows that, after partial true feature learning, the increment of signal component is mostly dominated by the noise component $\mathcal{V}_i^{(t)}$, which thus implies that the growth of signal component will converge when $\mathcal{V}_i^{(t)} = \Omega(1)$.

**Phase III: After that, by the quadratic increment of noise component (Lemma 4.4), the total noise $\mathcal{V}_i^{(t)}$ eventually attains the order $\Omega(1)$, which implies the model memorizes the spurious feature (data-wise noise) in final.**

**Lemma 4.4.** *(Lower Bound of Noise Component Growth) For each $i \in [N]$, $r \in [m]$ and $j \in [P] \setminus \text{signal}(\boldsymbol{X}_i)$ and any $t \geq 1$, the noise component grows as*

$$v_{i,j,r}^{(t)} \geq v_{i,j,r}^{(0)} + \Theta \left( \frac{\eta \sigma^2 d}{N} \right) \sum_{s=0}^{t-1} \psi(\mathcal{V}_i^{(s)}) \left( v_{i,j,r}^{(s)} \right)^2 - \tilde{O}(P \sigma^2 \alpha^{-1} \sqrt{d}).$$

The practical implication of Lemma 4.4 is two-fold. First, by the quadratic increment of noise component, we derive that, after $T_1 = \Theta \left( \frac{N}{\eta \sigma_0 \sigma^3 d^{\frac{3}{2}}} \right)$, the total noise memorization $\mathcal{V}_i^{(T)}$ attains the order $\Omega(1)$, which suggests that the model is able to robustly classify adversarial examples by memorizing the data-wise noise. Second, combined with Lemma 4.3, the maximum signal component $\max_{r \in [m]} u_r^{(t)}$ will maintain the order $\Theta(\alpha^{-1})$, which implies the conclusion of Theorem 4.1.

## 5 THE EFFICIENCY OF CGRO CLASSIFIER VIA REPRESENTATION COMPLEXITY

In this section, under more general data assumption, we provide a theoretical understanding of CGRO phenomenon from the view of representation complexity. We present the data assumption as follow.

**Assumption 5.1.** *Let $\mathcal{D}$ be a $D-$dimensional data distribution such that*

1. *(Well-Separated) The supporting set $\mathrm{supp}(\mathcal{D}) \in [0,1]^D$ can be divided into two disjoint set $\mathcal{A}$ and $\mathcal{B}$ that correspond to the positive and negative classes respectively. And it holds that $\mathrm{dist}_p(\mathcal{A}, \mathcal{B}) > 2\delta$, where $\delta$ is the $\ell_p$ adversarial perturbation radius.*

2. *(Neural-Separable) There exists a clean classifier that can be represented as a ReLU network with $\mathrm{poly}(D)$ parameters, which means that, under the distribution $\mathcal{D}$, the network achieves zero clean test error but its robust test error is at least $\Omega(1)$.*

Under Assumption 5.1, the above two conditions has been empirically observed in many practical works, which is also discussed in Section 1. Then, we have the following result about CGRO classifier.

**Theorem 5.2.** *(Polynomial Upper Bound for CGRO Classifier) Under Assumption 5.1, with $N-$sample training dataset $\mathcal{S} = \{(\boldsymbol{X}_1, y_1), (\boldsymbol{X}_2, y_2), \dots, (\boldsymbol{X}_N, y_N)\}$ drawn from the data distribution $\mathcal{D}$, there exists a CGRO classifier that can be represented as a ReLU network with $\mathrm{poly}(D) + \tilde{O}(ND)$ parameters, which means that, under the distribution $\mathcal{D}$ and dataset $\mathcal{S}$, the network achieves zero clean test and robust training errors but its robust test error is at least $\Omega(1)$.*

*Proof Sketch.* The proof idea of Theorem 5.2 is to approximate the classifier $f_{\mathcal{S}}$ proposed in (C) by ReLU network. Indeed, the function $f_{\mathcal{S}}$ can be rewritten as

$$f_{\mathcal{S}}(\boldsymbol{X}) = \underbrace{f_{\mathrm{clean}}(\boldsymbol{X})}_{\mathrm{poly}(D)} + \underbrace{\sum_{i=1}^{N}(y_i - f_{\mathrm{clean}}(\boldsymbol{X}))\mathbb{I}\{\|\boldsymbol{X} - \boldsymbol{X}_i\|_p \leq \delta\}}_{\text{weighted sum of robust local indicators}}.$$

Base on this, we use ReLU nets to approximate the distance function $d_i(\boldsymbol{X}) := \|\boldsymbol{X} - \boldsymbol{X}_i\|_p$ efficiently, and it is noticed that the exact indicator $\mathbb{I}\{\cdot\}$ can be approximated by a soft indicator that is represented by two ReLU neurons. Combined with two above results, we know there exists a ReLU net $f$ with at most $\mathrm{poly}(D) + \tilde{O}(ND)$ parameters such that $\|f - f_{\mathcal{S}}\|_{\ell_\infty([0,1]^D)} = o(1)$, which immediately implies Theorem 5.2. $\square$

However, to achieve robust generalization, higher complexity seems necessary. We generalize the conclusion in Li et al. (2022) from linear-separable assumption to neural-separable assumption.

**Theorem 5.3.** *(Exponential Lower Bound for Robust Classifer) Let $\mathcal{F}_M$ be the family of function represented by ReLU networks with at most $M$ parameters. There exists a number $M_D = \Omega(\exp(D))$ and a distribution $\mathcal{D}$ satisfying Assumption 5.1 such that, for any classifier in the family $\mathcal{F}_{M_D}$, under the distribution $\mathcal{D}$, the robust test error is at least $\Omega(1)$.*

Therefore, we derive the following inequalities,

$$\text{Representation Complexity: } \underbrace{Clean\ Classifier}_{\mathrm{poly}(D)} \lesssim \underbrace{CGRO\ Classifier}_{\mathrm{poly}(D)+\tilde{O}(ND)} \ll \underbrace{Robust\ Classifier}_{\Omega(\exp(D))}.$$

This inequalities states that while CGRO classifiers have mildly higher representation complexity than clean classifiers, adversarial robustness requires excessively higher complexity, which may lead adversarial training to converge to CGRO regime under the general data assumption.

## 6 EXPERIMENTS

In this section, we demonstrate that adversarial training converges to similarities of the construction $f_{\mathcal{S}}$ of (C) on real image datasets, which results in CGRO. In fact, we need to verify models trained by adversarial training tend to memorize data by approximating local robust indicators on training data.

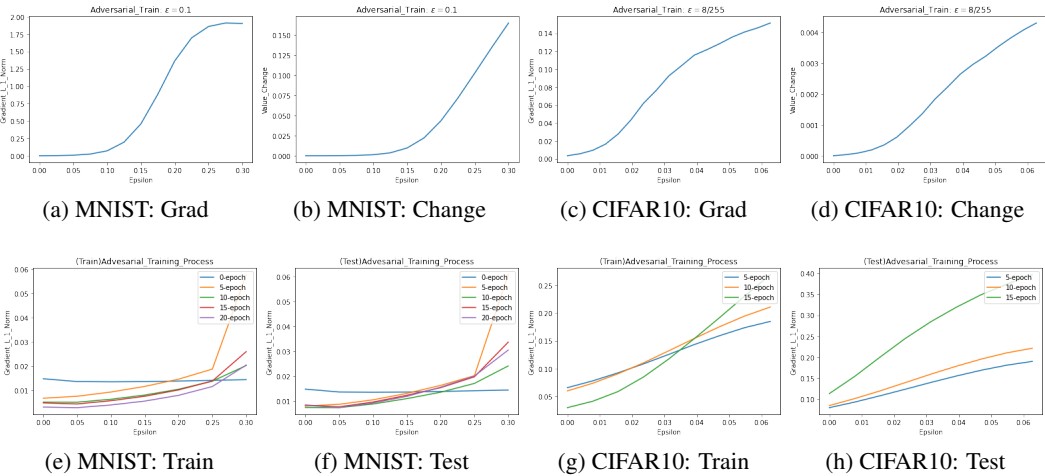

Figure 2: Experiment Results ($\ell_\infty$ Perturbation Radius $\epsilon_0 = 0.1$ on MNIST, $= 8/255$ on CIFAR10).

Concretely, for given loss $\mathcal{L}(\cdot, \cdot)$, instance $(\boldsymbol{X}, y)$ and model $f$, we use two measurements, maximum gradient norm within the neighborhood of training data, $\max_{\|\boldsymbol{\xi}\|_\infty \leq \delta} \|\nabla_{\boldsymbol{X}} \mathcal{L}(f(\boldsymbol{X} + \boldsymbol{\xi}), y)\|_1$ and maximum loss function value change $\max_{\|\boldsymbol{\xi}\|_\infty \leq \delta}[\mathcal{L}(f(\boldsymbol{X} + \boldsymbol{\xi}), y) - \mathcal{L}(f(\boldsymbol{X}), y)]$. The former measures the $\delta-$local flatness on $(\boldsymbol{X}, y)$, and the latter measures $\delta-$local adversarial robustness on $(\boldsymbol{X}, y)$, which both describe the key information of loss landscape over input.

**Experiment Settings.** In numerical experiments, we mainly focus on two common real-image datasets, MNIST and CIFAR10. During adversarial training, we use cyclical learning rates and mixed precision technique (Wong et al., 2020). On MNIST, we use a LeNet5 architecture and train total 20 epochs. On CIFAR10, we use a Resnet9 architecture and train total 15 epochs.

**Numerical Results.** First, we apply the adversarial training method to train models by a fixed perturbation radius $\epsilon_0$, and then we compute empirical average of maximum gradient norm and maximum loss change on training data within different perturbation radius $\epsilon$. We can see numerical results in Figure 2 (a∼d), and it shows that loss landscape has flatness within the training radius, but is very sharp outside, which practically demonstrates our conjecture on real image datasets.

**Learning Process.** We also focus on the dynamics of loss landscape over input during the adversarial learning process. Thus, we compute empirical average of maximum gradient norm within different perturbation radius $\epsilon$ and in different training epochs. The numerical results are plotted in Figure 2 (e∼h). Both on MNIST and CIFAR10, with epochs increasing, it is observed that the training curve descents within training perturbation radius, which implies models learn the local robust indicators to robustly memorize training data. However, the test curve of CIFAR10 ascents within training radius instead, which is consistent with our theoretical analysis in Section 4.

**Robust Generalization Bound.** Moreover, we prove a robust generalization bound based on *global flatness* of loss landscape (see in Appendix E). We show that, while adversarial training achieves local flatness by robust memorization, the model lacks global flatness, which causes robust overfitting.

# 7 CONCLUSION AND FUTURE WORK

In this paper, we present a theoretical understanding of clean generalization and robust overfitting (CGRO) phenomenon in adversarial training. Our main contribution is that, under our theoretical framework, we prove that neural network trained by adversarial training partially learns the true feature but memorizes the random noise in training data, which leads to CGRO phenomenon. In all, we believe that our work provides some theoretical insights into existing adversarial training methods. An important future work is to generalize our analysis of feature learning process to deep CNN models with other adversarial-example generative algorithms, such as FGSM and PGD attacks.

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

# A PRELIMINARIES

First, we present a technique called *Tensor Power Method* proposed by Allen-Zhu & Li (2020a;b).

**Lemma A.1.** *Let $\left\{z^{(t)}\right\}_{t=0}^{T}$ be a positive sequence defined by the following recursions*

$$\begin{cases} z^{(t+1)} \geq z^{(t)} + m\left(z^{(t)}\right)^2 \\ z^{(t+1)} \leq z^{(t)} + M\left(z^{(t)}\right)^2 \end{cases},$$

*where $z^{(0)} > 0$ is the initialization and $m, M > 0$. Let $v > 0$ such that $z^{(0)} \leq v$. Then, the time $t_0$ such that $z_t \geq v$ for all $t \geq t_0$ is:*

$$t_0 = \frac{3}{mz^{(0)}} + \frac{8M}{m}\left\lceil \frac{\log(v/z_0)}{\log(2)} \right\rceil.$$

**Lemma A.2.** *Let $\left\{z^{(t)}\right\}_{t=0}^{T}$ be a positive sequence defined by the following recursion*

$$\begin{cases} z^{(t)} \geq z^{(0)} + A\sum_{s=0}^{t-1}\left(z^{(s)}\right)^2 - C \\ z^{(t)} \leq z^{(0)} + A\sum_{s=0}^{t-1}\left(z^{(s)}\right)^2 + C \end{cases}$$

*where $A, C > 0$ and $z^{(0)} > 0$ is the initialization. Assume that $C \leq z^{(0)}/2$. Let $v > 0$ such that $z^{(0)} \leq v$. Then, the time $t$ such that $z^{(t)} \geq v$ is upper bounded as:*

$$t_0 = 8\left\lceil \frac{\log(v/z_0)}{\log(2)} \right\rceil + \frac{21}{\left(z^{(0)}\right)A}.$$

**Lemma A.3.** *Let $\mathcal{T} \geq 0$. Let $(z_t)_{t>\mathcal{T}}$ be a non-negative sequence that satisfies the recursion: $z^{(t+1)} \leq z^{(t)} - A\left(z^{(t)}\right)^2$, for $A > 0$. Then, it is bounded at a time $t > \mathcal{T}$ as*

$$z^{(t)} \leq \frac{1}{A(t-\mathcal{T})}.$$

Then, we provide a probability inequality proved by Jelassi & Li (2022).

**Lemma A.4.** *Let $\{\boldsymbol{v}_r\}_{r=1}^{m}$ be vectors in $\mathbb{R}^d$ such that there exist a unit norm vector $\boldsymbol{x}$ that satisfies $\left|\sum_{r=1}^{m}\langle \boldsymbol{v}_r, \boldsymbol{x}\rangle^3\right| \geq 1$. Then, for $\boldsymbol{\xi}_1, \ldots, \boldsymbol{\xi}_k \sim \mathcal{N}\left(0, \sigma^2\mathbf{I}_d\right)$ i.i.d., we have:*

$$\mathbb{P}\left[\left|\sum_{j=1}^{P}\sum_{r=1}^{m}\langle \boldsymbol{v}_r, \boldsymbol{\xi}_j\rangle^3\right| \geq \tilde{\Omega}\left(\sigma^3\right)\right] \geq 1 - \frac{O(d)}{2^{1/d}}.$$

Next, we introduce some concepts about learning theory.

**Definition A.5** (growth function). *Let $\mathcal{F}$ be a class of functions from $\mathcal{X} \subset \mathbb{R}^d$ to $\{-1, +1\}$. For any integer $m \geq 0$, we define the growth function of $\mathcal{F}$ to be*

$$\Pi_{\mathcal{F}}(m) = \max_{x_i \in \mathcal{X}, 1 \leq i \leq m} \left|\{(f(x_1), f(x_2), \cdots, f(x_m)) : f \in \mathcal{F}\}\right|.$$

*In particular, if $\left|\{(f(x_1), f(x_2), \cdots, f(x_m)) : f \in \mathcal{F}\}\right| = 2^m$, then $(x_1, x_2, \cdots, x_m)$ is said to be shattered by $\mathcal{F}$.*

**Definition A.6** (Vapnik-Chervonenkis dimension). *Let $\mathcal{F}$ be a class of functions from $\mathcal{X} \subset \mathbb{R}^D$ to $\{-1, +1\}$. The VC-dimension of $\mathcal{F}$, denoted by VC-dim($\mathcal{F}$), is defined as the largest integer $m \geq 0$ such that $\Pi_{\mathcal{F}}(m) = 2^m$. For real-value function class $\mathcal{H}$, we define VC-dim($\mathcal{H}$) := VC-dim($\mathrm{sgn}(\mathcal{H})$).*

The following result gives a nearly-tight upper bound on the VC-dimension of neural networks.

**Lemma A.7.** *(Bartlett et al., 2019, Theorem 6) Consider a ReLU network with $L$ layers and $W$ total parameters. Let $F$ be the set of (real-valued) functions computed by this network. Then we have VC-dim($F$) $= O(WL\log(W))$.*

The growth function is connected to the VC-dimension via the following lemma; see e.g. (Anthony et al., 1999, Theorem 7.6).

**Lemma A.8.** *Suppose that VC-dim($\mathcal{F}$) $= k$, then $\Pi_m(\mathcal{F}) \leq \sum_{i=0}^{k}\binom{m}{i}$. In particular, we have $\Pi_m(\mathcal{F}) \leq (em/k)^k$ for all $m > k + 1$.*

# B  FEATURE LEARNING

In this section, we provide a full introduction to feature learning, which is widely applied in theoretical works (Allen-Zhu & Li, 2020a;b; 2022; Shen et al., 2022; Jelassi & Li, 2022; Jelassi et al., 2022; Chen et al., 2022) explore what and how neural networks learn in different tasks. In this work, we first leverage feature learning theory to explain CGRO phenomenon in adversarial training. Specifically, for an arbitrary clean training data-point $(\boldsymbol{X}, y) \sim \mathcal{D}$ and a given model $f_{\boldsymbol{W}}$, we focus on

- **True Feature Learning.** We project the weight $\boldsymbol{W}$ on the meaningful signal vector to measure the correlation between the model and the true feature as

$$\mathcal{U} := \sum_{r=1}^{m} \langle \boldsymbol{w}_r, \boldsymbol{w}^* \rangle^3 .$$

- **Spurious Feature Learning.** We project the weight $\boldsymbol{W}$ on the random noise to measure the correlation between the model and the spurious feature as

$$\mathcal{V} := y \sum_{r=1}^{m} \sum_{j \in [P] \backslash \mathrm{signal}(\boldsymbol{X})} \langle \boldsymbol{w}_r, \boldsymbol{X}[j] \rangle^3 .$$

We then calculate the model's classification correctness on certain clean data point as

$$
\begin{aligned}
y f_{\boldsymbol{W}}(\boldsymbol{X}) &= y \sum_{r=1}^{m} \langle \boldsymbol{w}_r, \boldsymbol{X}[\mathrm{signal}(\boldsymbol{X})] \rangle^3 + y \sum_{r=1}^{m} \sum_{j \in [P] \backslash \mathrm{signal}(\boldsymbol{X})} \langle \boldsymbol{w}_r, \boldsymbol{X}[j] \rangle^3 \\
&= \underbrace{y \sum_{r=1}^{m} \langle \boldsymbol{w}_r, \alpha y \boldsymbol{w}^* \rangle^3}_{\alpha^3 \mathcal{U}} + \underbrace{y \sum_{r=1}^{m} \sum_{j \in [P] \backslash \mathrm{signal}(\boldsymbol{X})} \langle \boldsymbol{w}_r, \boldsymbol{X}[j] \rangle^3}_{\mathcal{V}} .
\end{aligned}
$$

Thus, the model correctly classify the data if and only if $\alpha^3 \mathcal{U} + \mathcal{V} \geq 0$, which holds in at least two cases. Indeed, one is that the model learns the true feature and ignores the spurious features, where $\mathcal{U} = \Omega(1) \gg |\mathcal{V}|$. Another is that the model doesn't learn the true feature but memorizes the spurious features, where $|\mathcal{U}| = o(1)$ and $|\mathcal{V}| = \Omega(1) \gg 0$.

Therefore, this analysis tells us that the model will generalize well for unseen data if the model learns true feature learning. But the model will overfit training data if the model only memorizes spurious features since the data-specific random noises are independent for distinct instances, which means that, with high probability, it holds that $\mathcal{V} = o(1)$ for unseen data $(\boldsymbol{X}, y)$.

We also calculate the model's classification correctness on perturbed data point, where we use geometry-inspired transferable attack proposed in (A) to generate adversarial example as

$$
\begin{aligned}
\boldsymbol{X}^{\mathrm{adv}}[j] &= \boldsymbol{X}[j] - \gamma \frac{g(\boldsymbol{X})}{\|\nabla_{\boldsymbol{X}} g(\boldsymbol{X})\|_2} \nabla_{\boldsymbol{X}} g(\boldsymbol{X})[j] \\
&= \boldsymbol{X}[j] - \gamma \frac{\langle \boldsymbol{w}^*, \boldsymbol{X}[\mathrm{signal}(\boldsymbol{X})] \rangle}{\|\boldsymbol{w}^*\|_2} \nabla_{\boldsymbol{X}} \langle \boldsymbol{w}^*, \boldsymbol{X}[\mathrm{signal}(\boldsymbol{X})] \rangle [j] \\
&= \begin{cases} \alpha(1 - \gamma) y \boldsymbol{w}^*, & j = \mathrm{signal}(\boldsymbol{X}) \\ \boldsymbol{X}[j], & j \in [p] \backslash \mathrm{signal}(\boldsymbol{X}) \end{cases}
\end{aligned}
$$

We then derive the correctness as

$$
\begin{aligned}
y f_{\boldsymbol{W}}(\boldsymbol{X}^{\mathrm{adv}}) &= y \sum_{r=1}^{m} \langle \boldsymbol{w}_r, \boldsymbol{X}^{\mathrm{adv}}[\mathrm{signal}(\boldsymbol{X})] \rangle^3 + y \sum_{r=1}^{m} \sum_{j \in [P] \backslash \mathrm{signal}(\boldsymbol{X})} \langle \boldsymbol{w}_r, \boldsymbol{X}^{\mathrm{adv}}[j] \rangle^3 \\
&= \underbrace{y \sum_{r=1}^{m} \langle \boldsymbol{w}_r, \alpha(1 - \gamma) y \boldsymbol{w}^* \rangle^3}_{\alpha^3 (1-\gamma)^3 \mathcal{U}} + \underbrace{y \sum_{r=1}^{m} \sum_{j \in [P] \backslash \mathrm{signal}(\boldsymbol{X})} \langle \boldsymbol{w}_r, \boldsymbol{X}[j] \rangle^3}_{\mathcal{V}} .
\end{aligned}
$$

Thus, the model correctly classify the perturbed data if and only if $\alpha^3 (1 - \gamma)^3 \mathcal{U} + \mathcal{V} \geq 0$, which implies that We can analyze the perturbed data similarly.

## C  PROOF FOR SECTION 4

In this section, we present the full proof for Section 4. First, we give detailed proofs of Lemma 4.2, Lemma 4.3 and Lemma 4.4. Then, we prove Theorem 4.1 base on the above lemmas.

We prove our main results using an induction. More specifically, we make the following assumptions for each iteration $t < T$.

**Hypothesis C.1.** *Throughout the learning process using the adversarial training update for $t < T$, we maintain that:*

- *(Uniform Bound for Signal Component) For each $r \in [m]$, we assume $u_r^{(t)} \leq \tilde{O}(\alpha^{-1})$.*

- *(Uniform Bound for Noise Component) For each $r \in [m]$, $i \in [N]$ and $j \in [P] \setminus \text{signal}(\boldsymbol{X}_i)$, we assume $|v_{i,j,r}^{(t)}| \leq \tilde{O}(1)$.*

In what follows, we assume these induction hypotheses for $t < T$ to prove our main results. We then prove these hypotheses for iteration $t = T$ in Lemma C.11.

Now, we first give proof details about Lemma 4.2.

**Theorem C.2.** *(Restatement of Lemma 4.2) For each $r \in [m]$ and any $t \geq 0$, the signal component grows as*

$$u_r^{(t+1)} \geq u_r^{(t)} + \Theta(\eta\alpha^3) \left(u_r^{(t)}\right)^2 \psi\left(\alpha^3 \sum_{k \in [m]} \left(u_k^{(t)}\right)^3\right),$$

*where we use $\psi(\cdot)$ to denote the negative sigmoid function $\psi(z) = \frac{1}{1+e^z}$ as well as Lemma 4.3,4.4.*

*Proof.* First, we calculate the gradient of adversarial loss with respect to $\boldsymbol{w}_r (r \in [m])$ as

$$\nabla_{\boldsymbol{w}_r}\widehat{\mathcal{L}}_{\text{adv}}(\boldsymbol{W}^{(t)}) = -\frac{3}{N}\sum_{i=1}^{N}\sum_{j=1}^{P}\left(\frac{(1-\lambda)y_i\left\langle\boldsymbol{w}_r^{(t)}, \boldsymbol{X}_i[j]\right\rangle^2}{1+\exp\left(y_i f_{\boldsymbol{W}^{(t)}}(\boldsymbol{X}_i)\right)}\boldsymbol{X}_i[j] + \frac{\lambda y_i\left\langle\boldsymbol{w}_r^{(t)}, \boldsymbol{X}_i^{\text{adv}}[j]\right\rangle^2}{1+\exp\left(y_i f_{\boldsymbol{W}^{(t)}}(\boldsymbol{X}_i^{\text{adv}})\right)}\boldsymbol{X}_i^{\text{adv}}[j]\right)$$

$$= -\frac{3}{N}\left(\left(u_r^{(t)}\right)^2\left(\sum_{i=1}^{N}(1-\lambda)\alpha^3\psi(y_i f_{\boldsymbol{W}^{(t)}}(\boldsymbol{X}_i)) + \lambda\alpha^3(1-\gamma)^3\psi(y_i f_{\boldsymbol{W}^{(t)}}(\boldsymbol{X}_i^{\text{adv}}))\right)\boldsymbol{w}^*\right.$$

$$\left. + \sum_{i=1}^{N}\sum_{j\neq\text{signal}(\boldsymbol{X}_i)}\left(v_{i,j,r}^{(t)}\right)^2\left((1-\lambda)\psi(y_i f_{\boldsymbol{W}^{(t)}}(\boldsymbol{X}_i)) + \lambda\psi(y_i f_{\boldsymbol{W}^{(t)}}(\boldsymbol{X}_i^{\text{adv}}))\right)\boldsymbol{X}_i[j]\right).$$

Then, we project the gradient descent algorithm equation $\boldsymbol{W}^{(t+1)} = \boldsymbol{W}^{(t)} - \eta\nabla_{\boldsymbol{W}}\widehat{\mathcal{L}}_{\text{adv}}\left(\boldsymbol{W}^{(t)}\right)$ on the signal vector $\boldsymbol{w}^*$. We derive the following result due to $\boldsymbol{X}_i[j] \perp \boldsymbol{w}^*$ for $j \in [P] \setminus \text{signal}(\boldsymbol{X}_i)$.

$$u_r^{(t+1)} = u_r^{(t)} + \frac{3\eta}{N}\left(u_r^{(t)}\right)^2\sum_{i=1}^{N}\left((1-\lambda)\alpha^3\psi(y_i f_{\boldsymbol{W}^{(t)}}(\boldsymbol{X}_i)) + \lambda\alpha^3(1-\gamma)^3\psi(y_i f_{\boldsymbol{W}^{(t)}}(\boldsymbol{X}_i^{\text{adv}}))\right)$$

$$\geq u_r^{(t)} + \frac{3\eta\alpha^3(1-\lambda)}{N}\left(u_r^{(t)}\right)^2\sum_{i=1}^{N}\psi(y_i f_{\boldsymbol{W}^{(t)}}(\boldsymbol{X}_i))$$

$$\geq u_r^{(t)} + \Theta(\eta\alpha^3)\left(u_r^{(t)}\right)^2\psi\left(\alpha^3\sum_{k\in[m]}\left(u_k^{(t)}\right)^3\right),$$

where we derive last inequality by using $\psi(y_i f_{\boldsymbol{W}^{(t)}}(\boldsymbol{X}_i)) = \Theta(1)\psi\left(\alpha^3\sum_{k\in[m]}\left(u_k^{(t)}\right)^3\right)$, which is obtained due to Hypothesis C.1.

$\square$

Consequently, we have the following result that shows the order of maximum signal component.

**Lemma C.3.** *During adversarial training, with high probability, it holds that, after $T_0 = \tilde{\Theta}\left(\frac{1}{\eta \alpha^3 \sigma_0}\right)$ iterations, for all $t \in [T_0, T]$, we have $\max_{r \in [m]} u_r^{(t)} \geq \tilde{\Omega}(\alpha^{-1})$.*

*Proof.* From the proof of Theorem C.2, we know that

$$u_r^{(t+1)} - u_r^{(t)} = \frac{3\eta}{N}\left(u_r^{(t)}\right)^2 \sum_{i=1}^{N}\left((1-\lambda)\alpha^3 \psi(y_i f_{\boldsymbol{W}^{(t)}}(\boldsymbol{X}_i)) + \lambda\alpha^3(1-\gamma)^3 \psi(y_i f_{\boldsymbol{W}^{(t)}}(\boldsymbol{X}_i^{\mathrm{adv}}))\right).$$

By applying Hypothesis C.1, we can simplify the above equation to the following inequalities.

$$\begin{cases} u_r^{(t+1)} \leq u_r^{(t)} + A\left(u_r^{(t)}\right)^2 \\ u_r^{(t+1)} \geq u_r^{(t)} + B\left(u_r^{(t)}\right)^2 \end{cases}$$

where $A$ and $B$ are respectively defined as:

$$A := \tilde{\Theta}(\eta)\left((1-\lambda)\alpha^3 + \lambda\alpha^3(1-\gamma)^3\right)$$
$$B := \tilde{\Theta}(\eta)(1-\lambda)\alpha^3.$$

At initialization, since we choose the weights $\boldsymbol{w}_r^{(0)} \sim \mathcal{N}\left(0, \sigma_0^2 \mathbf{I}_d\right)$, we know the initial signal components $u_r^{(0)}$ are i.i.d. zero-mean Gaussian random variables, which implies that the probability that at least one of the $u_r^{(0)}$ is non-negative is $1 - \left(\frac{1}{2}\right)^m = 1 - o(1)$.

Thus, with high probability, there exists an initial signal component $u_{r'}^{(0)} \geq 0$. By using Tensor Power Method (Lemma A.1) and setting $v = \tilde{\Theta}(\alpha^{-1})$, we have the threshold iteration $T_0$ as

$$T_0 = \frac{\tilde{\Theta}(1)}{\eta\alpha^3\sigma_0} + \frac{\tilde{\Theta}(1)\left((1-\lambda)\alpha^3 + \lambda\beta^3\right)}{(1-\lambda)\alpha^3}\left\lceil\frac{-\log\left(\tilde{\Theta}(\sigma_0\alpha)\right)}{\log(2)}\right\rceil.$$

$\square$

Next, we prove Lemma 4.3 to give an upper bound of signal components' growth.

**Theorem C.4.** *(Restatement of Lemma 4.3) For $T_0 = \Theta\left(\frac{1}{\eta\alpha^3\sigma_0}\right)$ and any $t \in [T_0, T]$, the signal component is upper bounded as*

$$\max_{r \in [m]} u_r^{(t)} \leq \tilde{O}(\alpha^{-1}) + \tilde{O}\left(\frac{\eta\alpha^3(1-\gamma)^3}{N}\right)\sum_{s=T_0}^{t-1}\sum_{i=1}^{N}\psi\left(\alpha^3(1-\gamma)^3\sum_{k \in [m]}\left(u_k^{(s)}\right)^3 + \mathcal{V}_i^{(s)}\right).$$

*Proof.* First, we analyze the upper bound of derivative generated by clean data. By following the proof of Theorem C.2, we know that, for each $r \in [m]$,

$$\max_{r \in [m]} u_r^{(t+1)} \geq \max_{r \in [m]} u_r^{(t)} + \frac{3\eta\alpha^3(1-\lambda)}{N}\left(\max_{r \in [m]} u_r^{(t)}\right)^2 \sum_{i=1}^{N}\psi(y_i f_{\boldsymbol{W}^{(t)}}(\boldsymbol{X}_i))$$

$$\geq \max_{r \in [m]} u_r^{(t)} + \tilde{\Omega}(\eta\alpha)\frac{1-\lambda}{N}\sum_{i=1}^{N}\psi(y_i f_{\boldsymbol{W}^{(t)}}(\boldsymbol{X}_i)),$$

where we obtain the first inequality by the definition of $\max_{r \in [m]} u_r^{(t)}, \max_{r \in [m]} u_r^{(t+1)}$, and we use $\max_{r \in [m]} u_r^{(t)} \geq \tilde{\Omega}(\alpha^{-1})$ derived by Lemma C.3 in the last inequality. Thus, we then have

$$\frac{1-\lambda}{N}\sum_{i=1}^{N}\psi(y_i f_{\boldsymbol{W}^{(t)}}(\boldsymbol{X}_i)) \leq \tilde{O}(\eta^{-1}\alpha^{-1})\left(\max_{r \in [m]} u_r^{(t+1)} - \max_{r \in [m]} u_r^{(t)}\right).$$

Now, we focus on $\max_{r \in [m]} u_r^{(t+1)} - \max_{r \in [m]} u_r^{(t)}$. By the non-decreasing property of $u_r^{(t)}$, we have

$$\max_{r \in [m]} u_r^{(t+1)} - \max_{r \in [m]} u_r^{(t)} \le \sum_{r \in [m]} \left( u_r^{(t+1)} - u_r^{(t)} \right)$$

$$\le (1-\lambda)\Theta(\eta\alpha)\psi\left(\alpha^3 \sum_{r \in [m]} \left(u_r^{(t)}\right)^3\right) \sum_{r \in [m]} \left(\alpha u_r^{(t)}\right)^2 + \lambda\Theta\left(\frac{\eta\alpha^3(1-\gamma)^3}{N}\right) \sum_{i=1}^{N} \psi(y_i f_{\boldsymbol{W}^{(t)}}(\boldsymbol{X}_i^{\mathrm{adv}}))$$

$$\le (1-\lambda)\tilde{O}(\eta\alpha^2)\phi\left(\alpha^3 \sum_{r \in [m]} \left(u_r^{(t)}\right)^3\right) + \lambda\Theta\left(\frac{\eta\alpha^3(1-\gamma)^3}{N}\right) \sum_{i=1}^{N} \psi(y_i f_{\boldsymbol{W}^{(t)}}(\boldsymbol{X}_i^{\mathrm{adv}})),$$

where we use $\phi(\cdot)$ to denote the logistics function defined as $\phi(z) = \log(1 + \exp(-z))$ and we derive the last inequality by Hypothesis C.1. Then, we know

$$\frac{1-\lambda}{N} \sum_{i=1}^{N} \psi(y_i f_{\boldsymbol{W}^{(t)}}(\boldsymbol{X}_i)) \le (1-\lambda)\tilde{O}(\alpha)\phi\left(\alpha^3 \sum_{r \in [m]} \left(u_r^{(t)}\right)^3\right)$$

$$+ \lambda\Theta\left(\frac{\alpha^2(1-\gamma)^3}{N}\right) \sum_{i=1}^{N} \psi(y_i f_{\boldsymbol{W}^{(t)}}(\boldsymbol{X}_i^{\mathrm{adv}})).$$

Then, we derive the following result by Hypothesis C.1 and the above inequality.

$$\max_{r \in [m]} u_r^{(t+1)} \le \max_{r \in [m]} u_r^{(t)} + \frac{3\eta}{N}\left(\max_{r \in [m]} u_r^{(t)}\right)^2 \sum_{i=1}^{N} \left((1-\lambda)\alpha^3\psi(y_i f_{\boldsymbol{W}^{(t)}}(\boldsymbol{X}_i))\right.$$

$$\left. + \lambda\alpha^3(1-\gamma)^3\psi(y_i f_{\boldsymbol{W}^{(t)}}(\boldsymbol{X}_i^{\mathrm{adv}}))\right)$$

$$\le \max_{r \in [m]} u_r^{(t)} + \tilde{\Theta}(\eta\alpha)\frac{1-\lambda}{N} \sum_{i=1}^{N} \psi(y_i f_{\boldsymbol{W}^{(t)}}(\boldsymbol{X}_i)) + \tilde{\Theta}\left(\frac{\eta\alpha(1-\gamma)^3}{N}\right) \sum_{i=1}^{N} \psi(y_i f_{\boldsymbol{W}^{(t)}}(\boldsymbol{X}_i^{\mathrm{adv}}))$$

$$\le \max_{r \in [m]} u_r^{(t)} + (1-\lambda)\tilde{O}(\eta\alpha^2)\phi\left(\alpha^3 \sum_{r \in [m]} \left(u_r^{(t)}\right)^3\right) + \tilde{\Theta}\left(\frac{\eta\alpha^3(1-\gamma)^3}{N}\right) \sum_{i=1}^{N} \psi(y_i f_{\boldsymbol{W}^{(t)}}(\boldsymbol{X}_i^{\mathrm{adv}}))$$

$$\le \max_{r \in [m]} u_r^{(t)} + \frac{(1-\lambda)\tilde{O}(\eta\alpha^2)}{1 + \exp\left(\alpha^3 \sum_{r \in [m]} \left(u_r^{(t)}\right)^3\right)} + \tilde{\Theta}\left(\frac{\eta\alpha^3(1-\gamma)^3}{N}\right) \sum_{i=1}^{N} \psi(y_i f_{\boldsymbol{W}^{(t)}}(\boldsymbol{X}_i^{\mathrm{adv}}))$$

$$\le \max_{r \in [m]} u_r^{(t)} + \frac{(1-\lambda)\tilde{O}(\eta\alpha^2)}{1 + \exp\left(\tilde{\Omega}(1)\right)} + \tilde{\Theta}\left(\frac{\eta\alpha^3(1-\gamma)^3}{N}\right) \sum_{i=1}^{N} \psi(y_i f_{\boldsymbol{W}^{(t)}}(\boldsymbol{X}_i^{\mathrm{adv}})).$$

By summing up iteration $s = T_0, \ldots, t-1$, we have the following result as

$$\max_{r \in [m]} u_r^{(t)} \le \max_{r \in [m]} u_r^{(T_0)} + \sum_{s=T_0}^{t-1} \frac{(1-\lambda)\tilde{O}(\eta\alpha^2)}{1 + \exp\left(\tilde{\Omega}(1)\right)} + \left(\frac{\eta\alpha^3(1-\gamma)^3}{N}\right) \sum_{s=T_0}^{t-1} \sum_{i=1}^{N} \psi(y_i f_{\boldsymbol{W}^{(s)}}(\boldsymbol{X}_i^{\mathrm{adv}}))$$

$$\le \tilde{O}(\alpha^{-1}) + \tilde{O}\left(\frac{\eta\alpha^3(1-\gamma)^3}{N}\right) \sum_{s=T_0}^{t-1} \sum_{i=1}^{N} \psi\left(\alpha^3(1-\gamma)^3 \sum_{k \in [m]} \left(u_k^{(s)}\right)^3 + \mathcal{V}_i^{(s)}\right).$$

Therefore, we derive the conclusion of Theorem C.4. $\qquad\square$

Next, we prove the following theorem about the update of noise components.

**Lemma C.5.** *For each $r \in [m]$, $i \in [N]$ and $j \in [P] \setminus \mathrm{signal}(\boldsymbol{X}_i)$, any iteration$_0, t$ such that $t_0 < t \leq T$, with high probability, it holds that*

$$\left| v_{i,j,r}^{(t)} - v_{i,j,r}^{(t_0)} - \Theta\left(\frac{\eta \sigma^2 d}{N}\right) \sum_{s=t_0}^{t-1} \tilde{\psi}_i^{(s)} \left(v_{i,j,r}^{(s)}\right)^2 \right| \leq \tilde{O}\left(\frac{\lambda \eta \alpha^3 (1-\gamma)^3}{N}\right) \sum_{s=t_0}^{t-1} \sum_{i=1}^{N} \psi(y_i f_{\boldsymbol{W}^{(s)}}(\boldsymbol{X}_i^{adv}))$$
$$+ \tilde{O}(P \sigma^2 \alpha^{-1} \sqrt{d}),$$

*where we use the notation $\tilde{\psi}_i^{(s)}$ to denote $(1-\lambda)\psi(y_i f_{\boldsymbol{W}^{(s)}}(\boldsymbol{X}_i)) + \lambda \psi(y_i f_{\boldsymbol{W}^{(s)}}(\boldsymbol{X}_i^{adv}))$.*

*Proof.* To obtain Lemma C.5, we prove the following stronger result by induction w.r.t. iteration $t$.

$$\left| v_{i,j,r}^{(t)} - v_{i,j,r}^{(t_0)} - \Theta\left(\frac{\eta \sigma^2 d}{N}\right) \sum_{s=t_0}^{t-1} \tilde{\psi}_i^{(s)} \left(v_{i,j,r}^{(s)}\right)^2 \right| \leq \tilde{O}(P \sigma^2 \alpha^{-1} \sqrt{d}) \left(1 + \lambda \alpha \eta + \frac{\alpha}{\sigma^2 d}\right)^{t-t_0-1} \sum_{q=0}^{t-t_0-1} (P^{-1}\sqrt{d})^{-q}$$
$$+ \tilde{O}\left(\frac{\lambda \eta \alpha^3 (1-\gamma)^3}{N}\right) \sum_{q=0}^{t-t_0-1} \sum_{s=t_0}^{t-q} \sum_{i=1}^{N} (P^{-1}\sqrt{d})^{-q} \psi(y_i f_{\boldsymbol{W}^{(s)}}(\boldsymbol{X}_i^{\mathrm{adv}})) \tag{1}$$

First, we project the training update on noise patch $\boldsymbol{X}_i[j]$ to verify the above inequality when $t = t_0 + 1$ as

$$\left| v_{i,j,r}^{(t_0+1)} - v_{i,j,r}^{(t_0)} - \Theta\left(\frac{\eta \sigma^2 d}{N}\right) \tilde{\psi}_i^{(t^0)} \left(v_{i,j,r}^{(s)}\right)^2 \right| \leq \Theta\left(\frac{\eta \sigma^2 d}{N}\right) \sum_{a=1}^{N} \sum_{b \neq \mathrm{signal}(\boldsymbol{X}_a)} \tilde{\psi}_a^{(t_0)} \left(v_{a,b,r}^{(t_0)}\right)^2$$
$$\leq \Theta(\eta P \sigma^2 \sqrt{d}) \frac{1-\lambda}{N} \sum_{i=1}^{N} \psi(y_i f_{\boldsymbol{W}^{(t_0)}}(\boldsymbol{X}_i))$$
$$+ \Theta(\eta P \sigma^2 \sqrt{d}) \frac{\lambda}{N} \sum_{i=1}^{N} \psi(y_i f_{\boldsymbol{W}^{(t_0)}}(\boldsymbol{X}_i^{\mathrm{adv}}))$$
$$\leq \tilde{O}(P \sigma^2 \alpha^{-1} \sqrt{d}) \left(1 + \lambda \alpha \eta + \frac{\alpha}{\sigma^2 d}\right),$$

where we apply $\frac{1-\lambda}{N} \sum_{i=1}^{N} \psi(y_i f_{\boldsymbol{W}^{(t_0)}}(\boldsymbol{X}_i)) \leq \tilde{O}(\eta^{-1}\alpha^{-1}) \left(\max_{r \in [m]} u_r^{(t_0+1)} - \max_{r \in [m]} u_r^{(t_0)}\right) \leq \tilde{O}(\eta^{-1}\alpha^{-2})$ and $\sum_{i=1}^{N} \psi(y_i f_{\boldsymbol{W}^{(t_0)}}(\boldsymbol{X}_i^{\mathrm{adv}})) \leq \tilde{O}(1)$ to derive the last inequality.

Next, we assume that the stronger result holds for iteration $t$, and then we prove the result for iteration $t+1$ as follow.

$$\left| v_{i,j,r}^{(t+1)} - v_{i,j,r}^{(t_0)} - \Theta\left(\frac{\eta \sigma^2 d}{N}\right) \sum_{s=t_0}^{t-1} \tilde{\psi}_i^{(s)} \left(v_{i,j,r}^{(s)}\right)^2 \right| \leq \Theta\left(\frac{\eta \sigma^2 d}{N}\right) \sum_{s=t_0}^{t-1} \sum_{a=1}^{N} \sum_{b \neq \mathrm{signal}(\boldsymbol{X}_a)} \tilde{\psi}_a^{(s)} \left(v_{a,b,r}^{(s)}\right)^2$$
$$+ \Theta(\eta P \sigma^2 \sqrt{d}) \frac{1-\lambda}{N} \sum_{i=1}^{N} \psi(y_i f_{\boldsymbol{W}^{(t)}}(\boldsymbol{X}_i)) + \Theta(\eta P \sigma^2 \sqrt{d}) \frac{\lambda}{N} \sum_{i=1}^{N} \psi(y_i f_{\boldsymbol{W}^{(t)}}(\boldsymbol{X}_i^{\mathrm{adv}})).$$

Then, we bound the first term in the right of the above inequality by our induction hypothesis for $t$, and we can derive

$$\left| v_{i,j,r}^{(t+1)} - v_{i,j,r}^{(t_0)} - \Theta\left(\frac{\eta \sigma^2 d}{N}\right) \sum_{s=t_0}^{t-1} \tilde{\psi}_i^{(s)} \left(v_{i,j,r}^{(s)}\right)^2 \right| \leq \tilde{O}(P \sigma^2 \alpha^{-1} \sqrt{d}) \left(1 + \lambda \alpha \eta + \frac{\alpha}{\sigma^2 d}\right)^{t-t_0-1} (P^{-1}\sqrt{d})^{-q}$$
$$+ \tilde{O}\left(\frac{\lambda \eta \alpha^3 (1-\gamma)^3}{N}\right) \sum_{q=0}^{t-t_0-1} \sum_{s=t_0}^{t-q} \sum_{i=1}^{N} (P^{-1}\sqrt{d})^{-q} \psi(y_i f_{\boldsymbol{W}^{(s)}}(\boldsymbol{X}_i^{\mathrm{adv}}))$$
$$+ \tilde{O}(P \sigma^2 \alpha^{-1} \sqrt{d}) \left(1 + \lambda \alpha \eta + \frac{\alpha}{\sigma^2 d}\right) + \Theta(\eta P \sigma^2 \sqrt{d}) \frac{\lambda}{N} \sum_{i=1}^{N} \psi(y_i f_{\boldsymbol{W}^{(t)}}(\boldsymbol{X}_i^{\mathrm{adv}})).$$

By summing up the terms, we proved the stronger result for $t + 1$.

Finally, we simplify the form of stronger result by using $\sum_{q=0}^{\infty}(P^{-1}\sqrt{d})^{-q} = (1 - P/\sqrt{d})^{-1} = \Theta(1)$, which implies the conclusion of Lemma C.5. $\qquad\square$

Now, we prove Lemma 4.4 based on Lemma C.5 as follow.

**Theorem C.6.** *(Restatement of Lemma 4.4) For each $i \in [N]$, $r \in [m]$ and $j \in [P] \setminus \mathrm{signal}(\boldsymbol{X}_i)$ and any $t \geq 1$, the signal component grows as*

$$v_{i,j,r}^{(t)} \geq v_{i,j,r}^{(0)} + \Theta\left(\frac{\eta\sigma^2 d}{N}\right)\sum_{s=0}^{t-1}\psi(\mathcal{V}_i^{(s)})\left(v_{i,j,r}^{(s)}\right)^2 - \tilde{O}(P\sigma^2\alpha^{-1}\sqrt{d}).$$

*Proof.* By applying the one-side inequality of Lemma C.5, we have

$$v_{i,j,r}^{(t)} - v_{i,j,r}^{(t_0)} - \Theta\left(\frac{\eta\sigma^2 d}{N}\right)\sum_{s=t_0}^{t-1}\tilde{\psi}_i^{(s)}\left(v_{i,j,r}^{(s)}\right)^2 \geq -\tilde{O}\left(\frac{\lambda\eta\alpha^3(1-\gamma)^3}{N}\right)\sum_{s=t_0}^{t-1}\sum_{i=1}^{N}\psi(y_i f_{\boldsymbol{W}^{(s)}}(\boldsymbol{X}_i^{\mathrm{adv}}))$$
$$- \tilde{O}(P\sigma^2\alpha^{-1}\sqrt{d}).$$

Thus, we obtain Theorem C.6 by using $\tilde{O}\left(\frac{\lambda\eta\alpha^3(1-\gamma)^3}{N}\right)\sum_{s=t_0}^{t-1}\sum_{i=1}^{N}\psi(y_i f_{\boldsymbol{W}^{(s)}}(\boldsymbol{X}_i^{\mathrm{adv}})) \leq \tilde{O}(\lambda\eta T\alpha^3(1-\gamma)^3) \leq \tilde{O}(P\sigma^2\alpha^{-1}\sqrt{d})$ and $\tilde{\psi}_i^{(s)} = \Theta(1)\psi(\mathcal{V}_i^{(s)})$ derived by Hypothesis C.1. $\qquad\square$

Consequently, we derive the upper bound of total noise components as follow.

**Lemma C.7.** *During adversarial training, with high probability, it holds that, after $T_1 = \Theta\left(\frac{N}{\eta\sigma_0\sigma^3 d^{\frac{3}{2}}}\right)$ iterations, for all $t \in [T_1, T]$ and each $i \in [N]$, we have $\mathcal{V}_i^{(t)} \geq \tilde{O}(1)$.*

*Proof.* By applying Lemma C.5 as the same in the proof of Theorem C.6, we know that

$$\left|v_{i,j,r}^{(t)} - v_{i,j,r}^{(t_0)} - \Theta\left(\frac{\eta\sigma^2 d}{N}\right)\sum_{s=t_0}^{t-1}\tilde{\psi}_i^{(s)}\left(v_{i,j,r}^{(s)}\right)^2\right| \leq \tilde{O}(P\sigma^2\alpha^{-1}\sqrt{d}),$$

which implies that, for any iteration $t \leq T$, we have

$$\begin{cases} v_{i,j,r}^{(t)} \geq v_{i,j,r}^{(0)} + A\sum_{s=0}^{t-1}\left(v_{i,j,r}^{(s)}\right)^2 - C \\ v_{i,j,r}^{(t)} \leq v_{i,j,r}^{(0)} + A\sum_{s=0}^{t-1}\left(v_{i,j,r}^{(s)}\right)^2 + C \end{cases},$$

where $A, C > 0$ are constants defined as

$$A = \frac{\tilde{\Theta}\left(\eta\sigma^2 d\right)}{N}, \quad C = \tilde{O}(P\sigma^2\alpha^{-1}\sqrt{d}).$$

At initialization, since we choose the weights $\boldsymbol{w}_r^{(0)} \sim \mathcal{N}\left(0, \sigma_0^2\mathbf{I}_d\right)$ and $\boldsymbol{X}_i[j] \sim \mathcal{N}\left(0, \sigma^2\mathbf{I}_d\right)$, we know the initial noise components $v_{i,j,r}^{(0)}$ are i.i.d. zero-mean Gaussian random variables, which implies that, with high probability, there exists at least one index $r'$ such that $v_{i,j,r}^{(0)} \geq \tilde{\Omega}(P\sigma^2\alpha^{-1}\sqrt{d})$. By using Tensor Power Method (Lemma A.2) and setting $v = \tilde{\Theta}(1)$, we have the threshold iteration $T_1$ as

$$T_1 = \frac{21N}{\tilde{\Theta}\left(\eta\sigma^2 d\right)v_{i,j,r}^{(0)}} + \frac{8N}{\tilde{\Theta}\left(\eta\sigma^2 d\right)\left(v_{i,j,r}^{(0)}\right)}\left\lceil\frac{\log\left(\frac{\tilde{O}(1)}{v_{i,j,r}^{(0)}}\right)}{\log(2)}\right\rceil.$$

Therefore, we get $T_1 = \Theta\left(\frac{N}{\eta\sigma_0\sigma^3 d^{\frac{3}{2}}}\right)$, and we use $\mathcal{V}_i^{(t)} = \sum_{r\in[m]}\sum_{j\in[P]\setminus\mathrm{signal}(\boldsymbol{X}_i)}\left(v_{i,j,r}^{(t)}\right)^3$ to derive $\mathcal{V}_i^{(t)} \geq \tilde{\Omega}(1)$. $\qquad\square$

Indeed, our aimed loss function $\widehat{\mathcal{L}}_{\mathrm{adv}}(\boldsymbol{W})$ is non-convex due to the non-linearity of our CNN model $f_{\boldsymbol{W}}$. To analyze the convergence of gradient algorithm, we need to prove the following condition that is used to show non-convexly global convergence (Karimi et al., 2016; Li et al., 2019).

**Lemma C.8.** *(Lojasiewicz Inequality for Non-convex Optimization) During adversarial training, with high probability, it holds that, after $T_1 = \Theta\left(\frac{N}{\eta\sigma_0\sigma^3 d^{\frac{3}{2}}}\right)$ iterations, for all $t \in [T_1, T]$, we have*

$$\left\|\nabla_{\boldsymbol{W}}\widehat{\mathcal{L}}_{adv}\left(\boldsymbol{W}^{(t)}\right)\right\|_2 \geq \tilde{\Omega}(1)\widehat{\mathcal{L}}_{adv}\left(\boldsymbol{W}^{(t)}\right).$$

*Proof.* To prove Lojasiewicz Inequality, we first recall the gradient w.r.t. $\boldsymbol{w}_r$ as

$$\nabla_{\boldsymbol{w}_r}\widehat{\mathcal{L}}_{\mathrm{adv}}(\boldsymbol{W}^{(t)}) = -\frac{3}{N}\left(\left(u_r^{(t)}\right)^2\left(\sum_{i=1}^{N}(1-\lambda)\alpha^3\psi(y_i f_{\boldsymbol{W}^{(t)}}(\boldsymbol{X}_i)) + \lambda\alpha^3(1-\gamma)^3\psi(y_i f_{\boldsymbol{W}^{(t)}}(\boldsymbol{X}_i^{\mathrm{adv}}))\right)\boldsymbol{w}^*\right.$$

$$\left.+ \sum_{i=1}^{N}\sum_{j\neq\mathrm{signal}(\boldsymbol{X}_i)}\left(v_{i,j,r}^{(t)}\right)^2\left((1-\lambda)\psi(y_i f_{\boldsymbol{W}^{(t)}}(\boldsymbol{X}_i)) + \lambda\psi(y_i f_{\boldsymbol{W}^{(t)}}(\boldsymbol{X}_i^{\mathrm{adv}}))\right)\boldsymbol{X}_i[j]\right).$$

Then, we project the gradient on the signal direction and total noise, respectively.

For the signal component, we have

$$\left\|\nabla_{\boldsymbol{w}_r}\widehat{\mathcal{L}}_{\mathrm{adv}}(\boldsymbol{W}^{(t)})\right\|_2^2 \geq \left\langle\nabla_{\boldsymbol{w}_r}\widehat{\mathcal{L}}_{\mathrm{adv}}(\boldsymbol{W}^{(t)}), \boldsymbol{w}^*\right\rangle^2$$

$$\geq \tilde{\Omega}(1)\left((1-\lambda)\alpha^3\left(u_r^{(t)}\right)^2\psi\left(\alpha^3\sum_{k\in[m]}\left(u_k^{(t)}\right)^3\right)\right)^2.$$

For the total noise component, we have

$$\left\|\nabla_{\boldsymbol{w}_r}\widehat{\mathcal{L}}_{\mathrm{adv}}(\boldsymbol{W}^{(t)})\right\|_2^2 \geq \left\langle\nabla_{\boldsymbol{w}_r}\widehat{\mathcal{L}}_{\mathrm{adv}}(\boldsymbol{W}^{(t)}), \frac{\sum_{i=1}^{N}\sum_{j\neq\mathrm{signal}(\boldsymbol{X}_i)}\boldsymbol{X}_i[j]}{\left\|\sum_{i=1}^{N}\sum_{j\neq\mathrm{signal}(\boldsymbol{X}_i)}\boldsymbol{X}_i[j]\right\|_2}\right\rangle^2$$

$$= \left\langle-\frac{3}{N}\sum_{i=1}^{N}\sum_{j\neq\mathrm{signal}(\boldsymbol{X}_i)}\left(v_{i,j,r}^{(t)}\right)^2\left((1-\lambda)\psi(y_i f_{\boldsymbol{W}^{(t)}}(\boldsymbol{X}_i)) + \lambda\psi(y_i f_{\boldsymbol{W}^{(t)}}(\boldsymbol{X}_i^{\mathrm{adv}}))\right)\boldsymbol{X}_i[j],\right.$$

$$\left.\frac{\sum_{a=1}^{N}\sum_{b\neq\mathrm{signal}(\boldsymbol{X}_a)}\boldsymbol{X}_a[b]}{\left\|\sum_{a=1}^{N}\sum_{b\neq\mathrm{signal}(\boldsymbol{X}_a)}\boldsymbol{X}_a[b]\right\|_2}\right\rangle^2$$

$$= \left(\left(\left\langle-\frac{3}{N}\sum_{i=1}^{N}\sum_{j\neq\mathrm{signal}(\boldsymbol{X}_i)}\left(v_{i,j,r}^{(t)}\right)^2(1-\lambda)\psi(y_i f_{\boldsymbol{W}^{(t)}}(\boldsymbol{X}_i))\boldsymbol{X}_i[j], \frac{\sum_{a=1}^{N}\sum_{b\neq\mathrm{signal}(\boldsymbol{X}_a)}\boldsymbol{X}_a[b]}{\left\|\sum_{a=1}^{N}\sum_{b\neq\mathrm{signal}(\boldsymbol{X}_a)}\boldsymbol{X}_a[b]\right\|_2}\right\rangle\right.\right.$$

$$\left.\left.+ \left\langle-\frac{3}{N}\sum_{i=1}^{N}\sum_{j\neq\mathrm{signal}(\boldsymbol{X}_i)}\left(v_{i,j,r}^{(t)}\right)^2\lambda\psi(y_i f_{\boldsymbol{W}^{(t)}}(\boldsymbol{X}_i^{\mathrm{adv}}))\boldsymbol{X}_i[j], \frac{\sum_{a=1}^{N}\sum_{b\neq\mathrm{signal}(\boldsymbol{X}_a)}\boldsymbol{X}_a[b]}{\left\|\sum_{a=1}^{N}\sum_{b\neq\mathrm{signal}(\boldsymbol{X}_a)}\boldsymbol{X}_a[b]\right\|_2}\right\rangle\right)^2\right).$$

For the first term, with high probability, it holds that

$$\left\langle-\frac{3}{N}\sum_{i=1}^{N}\sum_{j\neq\mathrm{signal}(\boldsymbol{X}_i)}\left(v_{i,j,r}^{(t)}\right)^2(1-\lambda)\psi(y_i f_{\boldsymbol{W}^{(t)}}(\boldsymbol{X}_i))\boldsymbol{X}_i[j], \frac{\sum_{a=1}^{N}\sum_{b\neq\mathrm{signal}(\boldsymbol{X}_a)}\boldsymbol{X}_a[b]}{\left\|\sum_{a=1}^{N}\sum_{b\neq\mathrm{signal}(\boldsymbol{X}_a)}\boldsymbol{X}_a[b]\right\|_2}\right\rangle$$

$$\geq -\frac{\tilde{O}(\sigma)(1-\lambda)}{N}\sum_{i=1}^{N}\sum_{j\neq\mathrm{signal}(\boldsymbol{X}_i)}\left(v_{i,j,r}^{(t)}\right)^2(1-\lambda)\psi(y_i f_{\boldsymbol{W}^{(t)}}(\boldsymbol{X}_i)),$$

where we use that $\left\langle \boldsymbol{X}_i[j], \frac{\sum_{a=1}^{N} \sum_{b \neq \text{signal}(\boldsymbol{X}_a)} \boldsymbol{X}_a[b]}{\left\|\sum_{a=1}^{N} \sum_{b \neq \text{signal}(\boldsymbol{X}_a)} \boldsymbol{X}_a[b]\right\|_2} \right\rangle$ is a sub-Gaussian random variable of pa-

rameter $\sigma$, which implies w.h.p. $\left| \left\langle \boldsymbol{X}_i[j], \frac{\sum_{a=1}^{N} \sum_{b \neq \text{signal}(\boldsymbol{X}_a)} \boldsymbol{X}_a[b]}{\left\|\sum_{a=1}^{N} \sum_{b \neq \text{signal}(\boldsymbol{X}_a)} \boldsymbol{X}_a[b]\right\|_2} \right\rangle \right| \leq \tilde{O}(\sigma).$

For the second term, with high probability, it holds that

$$\left\langle \frac{3}{N} \sum_{i=1}^{N} \sum_{j \neq \text{signal}(\boldsymbol{X}_i)} \left(v_{i,j,r}^{(t)}\right)^2 \lambda \psi(y_i f_{\boldsymbol{W}^{(t)}}(\boldsymbol{X}_i^{\text{adv}})) \boldsymbol{X}_i[j], \frac{\sum_{a=1}^{N} \sum_{b \neq \text{signal}(\boldsymbol{X}_a)} \boldsymbol{X}_a[b]}{\left\|\sum_{a=1}^{N} \sum_{b \neq \text{signal}(\boldsymbol{X}_a)} \boldsymbol{X}_a[b]\right\|_2} \right\rangle$$

$$= \frac{\Theta(1)}{N} \sum_{i=1}^{N} \sum_{j \neq \text{signal}(\boldsymbol{X}_i)} \left(v_{i,j,r}^{(t)}\right)^2 \lambda \psi(y_i f_{\boldsymbol{W}^{(t)}}(\boldsymbol{X}_i^{\text{adv}})) \frac{\|\boldsymbol{X}_i[j]\|_2^2}{\left\|\sum_{a=1}^{N} \sum_{b \neq \text{signal}(\boldsymbol{X}_a)} \boldsymbol{X}_a[b]\right\|_2}$$

$$= \frac{\Theta(\sigma \sqrt{d})}{N} \sum_{i=1}^{N} \sum_{j \neq \text{signal}(\boldsymbol{X}_i)} \left(v_{i,j,r}^{(t)}\right)^2 \lambda \psi(y_i f_{\boldsymbol{W}^{(t)}}(\boldsymbol{X}_i^{\text{adv}})),$$

where we use w.h.p. $\frac{\langle \boldsymbol{X}_i[j], \boldsymbol{X}_{i'}[j'] \rangle}{\left\|\sum_{a=1}^{N} \sum_{b \neq \text{signal}(\boldsymbol{X}_a)} \boldsymbol{X}_a[b]\right\|_2} \leq \frac{\Theta\left(\frac{1}{\sqrt{d}}\right) \|\boldsymbol{X}_i[j]\|_2^2}{\left\|\sum_{a=1}^{N} \sum_{b \neq \text{signal}(\boldsymbol{X}_a)} \boldsymbol{X}_a[b]\right\|_2}$ for $(i,j) \neq (i',j')$.

Now, combine the above bounds, we derive

$$\sum_{r=1}^{m} \left\|\nabla_{\boldsymbol{w}_r} \widehat{\mathcal{L}}_{\text{adv}}(\boldsymbol{W}^{(t)})\right\|_2^2 \geq \sum_{r=1}^{m} \left\langle \nabla_{\boldsymbol{w}_r} \widehat{\mathcal{L}}_{\text{adv}}(\boldsymbol{W}^{(t)}), \boldsymbol{w}^* \right\rangle^2$$

$$+ \sum_{r=1}^{m} \left\langle \nabla_{\boldsymbol{w}_r} \widehat{\mathcal{L}}_{\text{adv}}(\boldsymbol{W}^{(t)}), \frac{\sum_{i=1}^{N} \sum_{j \neq \text{signal}(\boldsymbol{X}_i)} \boldsymbol{X}_i[j]}{\left\|\sum_{i=1}^{N} \sum_{j \neq \text{signal}(\boldsymbol{X}_i)} \boldsymbol{X}_i[j]\right\|_2} \right\rangle^2$$

$$\geq \Omega\left(\frac{1}{m}\right) \left((1-\lambda)\alpha^3 \sum_{r=1}^{m} \left(u_r^{(t)}\right)^2 \psi\left(\alpha^3 \sum_{k \in [m]} \left(u_k^{(t)}\right)^3\right)\right.$$

$$+ \frac{\Theta(\sigma \sqrt{d})}{N} \sum_{r=1}^{m} \sum_{i=1}^{N} \sum_{j \neq \text{signal}(\boldsymbol{X}_i)} \left(v_{i,j,r}^{(t)}\right)^2 \lambda \psi(y_i f_{\boldsymbol{W}^{(t)}}(\boldsymbol{X}_i^{\text{adv}}))$$

$$\left. - \frac{\tilde{O}(\sigma)(1-\lambda)}{N} \sum_{r=1}^{m} \sum_{i=1}^{N} \sum_{j \neq \text{signal}(\boldsymbol{X}_i)} \left(v_{i,j,r}^{(t)}\right)^2 (1-\lambda) \psi(y_i f_{\boldsymbol{W}^{(t)}}(\boldsymbol{X}_i))\right)^2$$

$$\geq \tilde{\Omega}(1) \left((1-\lambda)\phi\left(\alpha^3 \sum_{r \in [m]} \left(u_r^{(t)}\right)^3\right) + \frac{\lambda}{N} \sum_{i=1}^{N} \phi\left(\mathcal{V}_i^{(t)}\right)\right)^2$$

$$\geq \tilde{\Omega}(1) \left(\widehat{\mathcal{L}}_{\text{adv}}\left(\boldsymbol{W}^{(t)}\right)\right)^2.$$

$\square$

Consequently, we derive the following sub-linear convergence result by applying Lojasiewicz Inequality.

**Lemma C.9.** *(Sub-linear Convergence for Adversarial Training) During adversarial training, with high probability, it holds that, after $T_1 = \Theta\left(\frac{N}{\eta \sigma_0 \alpha^3 d^{\frac{3}{2}}}\right)$ iterations, the adversarial training loss sub-linearly converges to zero as*

$$\widehat{\mathcal{L}}_{adv}\left(\boldsymbol{W}^{(t)}\right) \leq \frac{\tilde{O}(1)}{\eta(t - T_1 + 1)}.$$

*Proof.* Due to the smoothness of loss function $\widehat{\mathcal{L}}_{\text{adv}}(\boldsymbol{W})$ and learning rate $\eta = \tilde{O}(1)$, we have

$$
\widehat{\mathcal{L}}_{\text{adv}}\left(\boldsymbol{W}^{(t+1)}\right) \leq \widehat{\mathcal{L}}_{\text{adv}}\left(\boldsymbol{W}^{(t)}\right) - \frac{\eta}{2}\left\|\nabla_{\boldsymbol{W}}\widehat{\mathcal{L}}_{\text{adv}}\left(\boldsymbol{W}^{(t)}\right)\right\|_2
$$
$$
\leq \widehat{\mathcal{L}}_{\text{adv}}\left(\boldsymbol{W}^{(t)}\right) - \tilde{\Omega}(\eta)\left(\widehat{\mathcal{L}}_{\text{adv}}\left(\boldsymbol{W}^{(t)}\right)\right)^2,
$$

where we use Lojasiewicz Inequality in the last inequality. Then, by applying Tensor Power Method (Lemma A.3), we obtain the sub-linear convergence rate. $\qquad\square$

Now, we present the following result to bound the derivative generated by training-adversarial examples.

**Lemma C.10.** *During adversarial training, with high probability, it holds that, after $T_1 = \Theta\left(\frac{N}{\eta\sigma_0\sigma^3 d^{\frac{3}{2}}}\right)$ iterations, we have $\frac{\lambda}{N}\sum_{s=0}^{t}\sum_{i=1}^{N}\psi(y_i f_{\boldsymbol{W}^{(s)}}(\boldsymbol{X}_i^{adv})) \leq \tilde{O}(\eta^{-1}\sigma_0^{-1}).$*

*Proof.* First, we bound the total derivative during iteration $s = T_1, \ldots, t$. By applying the conclusion of Lemma C.5, we have

$$
\frac{\lambda}{N}\sum_{s=T_1}^{t}\sum_{i=1}^{N}\psi(y_i f_{\boldsymbol{W}^{(s)}}(\boldsymbol{X}_i^{\text{adv}})) \leq \frac{\tilde{O}(1)}{N}\sum_{s=T_1}^{t-1}\sum_{i=1}^{N}\tilde{\psi}_i^{(s)}\left(v_{i,j,r}^{(s)}\right)^2
$$
$$
+ \tilde{O}\left(\frac{\lambda\alpha^3(1-\gamma)^3}{N\sigma^2 d}\right)\sum_{s=T_1}^{t-1}\sum_{i=1}^{N}\psi(y_i f_{\boldsymbol{W}^{(s)}}(\boldsymbol{X}_i^{\text{adv}})) + \tilde{O}\left(\frac{P}{\eta\alpha\sqrt{d}}\right).
$$

Due to $\tilde{O}\left(\frac{\alpha^3(1-\gamma)^3}{\sigma^2 d}\right) \ll 1$, we know

$$
\frac{\lambda}{N}\sum_{s=T_1}^{t}\sum_{i=1}^{N}\psi(y_i f_{\boldsymbol{W}^{(s)}}(\boldsymbol{X}_i^{\text{adv}})) \leq \frac{\tilde{O}(1)}{N}\sum_{s=T_1}^{t-1}\sum_{i=1}^{N}\tilde{\psi}_i^{(s)}\left(v_{i,j,r}^{(s)}\right)^2 + \tilde{O}\left(\frac{P}{\eta\alpha\sqrt{d}}\right)
$$
$$
\leq \frac{\tilde{O}(1)}{N}\sum_{s=T_1}^{t-1}\sum_{i=1}^{N}\phi\left(\mathcal{V}_i^{(s)}\right) + \tilde{O}\left(\frac{P}{\eta\alpha\sqrt{d}}\right)
$$
$$
\leq \tilde{O}(1)\sum_{s=T_1}^{t-1}\widehat{\mathcal{L}}_{\text{adv}}\left(\boldsymbol{W}^{(t)}\right) + \tilde{O}\left(\frac{P}{\eta\alpha\sqrt{d}}\right)
$$
$$
\leq \tilde{O}(1)\sum_{s=T_1}^{t-1}\frac{\tilde{O}(1)}{\eta(t-T_1+1)} + \tilde{O}\left(\frac{P}{\eta\alpha\sqrt{d}}\right) \leq \tilde{O}(\eta^{-1}).
$$

Thus, we obtain $\frac{\lambda}{N}\sum_{s=0}^{t}\sum_{i=1}^{N}\psi(y_i f_{\boldsymbol{W}^{(s)}}(\boldsymbol{X}_i^{\text{adv}})) \leq \tilde{O}(\sigma_0^{-1}) + \tilde{O}(\eta^{-1}) \leq \tilde{O}(\eta^{-1}\sigma_0^{-1})$. $\qquad\square$

Consequently, we have the following lemma that verifies Hypothesis C.1 for $t = T$.

**Lemma C.11.** *During adversarial training, with high probability, it holds that, for any $t \leq T$, we have $\max_{r\in[m]} u_r^{(t)} \leq \tilde{O}(\alpha^{-1})$ and $|v_{i,j,r}^{(t)}| \leq \tilde{O}(1)$ for each $r \in [m], i \in [N], j \in [P] \setminus \text{signal}(\boldsymbol{X}_i).$*

*Proof.* Combined with Theorem C.4 and Lemma C.10, we can derive $\max_{r\in[m]} u_r^{(T)} \leq \tilde{O}(\alpha^{-1})$.

By applying Lemma C.5, we have

$$
|v_{i,j,r}^{(T)}| \leq |v_{i,j,r}^{(0)}| + \Theta\left(\frac{\eta\sigma^2 d}{N}\right)\sum_{s=t_0}^{t-1}\tilde{\psi}_i^{(s)}\left(v_{i,j,r}^{(s)}\right)^2
$$
$$
+ \tilde{O}\left(\frac{\lambda\eta\alpha^3(1-\gamma)^3}{N}\right)\sum_{s=t_0}^{t-1}\sum_{i=1}^{N}\psi(y_i f_{\boldsymbol{W}^{(s)}}(\boldsymbol{X}_i^{\text{adv}})) + \tilde{O}(P\sigma^2\alpha^{-1}\sqrt{d})
$$
$$
\leq \tilde{O}(1) + \tilde{O}(\sigma^2 d) + \tilde{O}(\alpha^3(1-\gamma)^3\sigma_0^{-1}) + \tilde{O}(P\sigma^2\alpha^{-1}\sqrt{d}) \leq \tilde{O}(1).
$$

Therefore, our Hypothesis C.1 holds for iteration $t = T$. $\qquad\square$

Finally, we prove our main result as follow.

**Theorem C.12.** *(Restatement of Theorem 4.1) Under Parameterization 3.1, we run the adversarial training algorithm to update the weight of the simplified CNN model for $T = \Omega(\text{poly}(d))$ iterations. Then, with high probability, it holds that the CNN model*

1. *partially learns the true feature, i.e. $\mathcal{U}^{(T)} = \Theta(\alpha^{-3})$;*

2. *exactly memorizes the spurious feature, i.e. for each $i \in [N], \mathcal{V}_i^{(T)} = \Theta(1)$,*

*where $\mathcal{U}^{(t)}$ and $\mathcal{V}_i^{(t)}$ is defined for $i-th$ instance $(\mathbf{X}_i, y_i)$ and $t-th$ iteration as the same in (1)(1). Consequently, the clean test error and robust training error are both smaller than $o(1)$, but the robust test error is at least $\frac{1}{2} - o(1)$.*

*Proof.* First, by applying Lemma C.3, Lemma C.7 and Lemma C.11, we know for any $i \in [N]$

$$\mathcal{U}^{(T)} = \sum_{r \in [m]} \left( u_r^{(T)} \right)^3 = \Theta(\alpha^{-3})$$

$$\mathcal{V}_i^{(T)} = \sum_{r \in [m]} \sum_{j \neq \text{signal}(\mathbf{X}_i)} \left( v_{i,j,r}^{(T)} \right)^3 = \Theta(1).$$

Then, since adversarial loss sub-linearly converges to zero i.e. $\widehat{\mathcal{L}}_{\text{adv}} \left( \mathbf{W}^{(T)} \right) \leq \frac{\tilde{O}(1)}{\eta(T - T_1 + 1)} \leq \tilde{O} \left( \frac{1}{\text{poly}(d)} \right) = o(1)$, the robust training error is also at most $o(1)$.

To analyze test errors, we decompose $\mathbf{w}_r^{(T)}$ into $\mathbf{w}_r^{(T)} = \mu_r^{(T)} \mathbf{w}^* + \boldsymbol{\beta}_r^{(T)}$ for each $r \in [m]$, where $\boldsymbol{\beta}_r^{(T)} \in (\text{span}(\mathbf{w}^*))^\perp$. Due to $\mathcal{V}_i^{(T)} = \Theta(1)$, we know $\|\boldsymbol{\beta}_r^{(T)}\|_2 = \Theta(1)$.

For the clean test error, we have

$$\mathbb{P}_{(\mathbf{X},y)\sim\mathcal{D}} \left[ y f_{\mathbf{W}^{(T)}}(\mathbf{X}) < 0 \right] = \mathbb{P}_{(\mathbf{X},y)\sim\mathcal{D}} \left[ \alpha^3 \sum_{r=1}^m \left( u_r^{(T)} \right)^3 + y \sum_{r=1}^m \sum_{j \in [P]\backslash\text{signal}(\mathbf{X})} \left\langle \mathbf{w}_r^{(T)}, \mathbf{X}[j] \right\rangle^3 < 0 \right]$$

$$\leq \mathbb{P}_{(\mathbf{X},y)\sim\mathcal{D}} \left[ \sum_{r=1}^m \sum_{j \in [P]\backslash\text{signal}(\mathbf{X})} \left\langle \boldsymbol{\beta}_r^{(T)}, \mathbf{X}[j] \right\rangle^3 \geq \tilde{\Omega}(1) \right]$$

$$\leq \exp \left( -\frac{\tilde{\Omega}(1)}{\sigma^6 \sum_{r=1}^m \|\boldsymbol{\beta}_r^{(T)}\|_2^6} \right) \leq O \left( \frac{1}{\text{poly}(d)} \right) = o(1),$$

where we use the fact that $\sum_{r=1}^m \sum_{j \in [P]\backslash\text{signal}(\mathbf{X})} \left\langle \boldsymbol{\beta}_r^{(T)}, \mathbf{X}[j] \right\rangle^3$ is a sub-Gaussian random variable with parameter $\sigma^3 \sqrt{(P-1) \sum_{r=1}^m \|\boldsymbol{\beta}_r^{(T)}\|_2^6}$.

For the robust test error, we use $\mathcal{A}(\cdot)$ to denote geometry-inspired transferable attack (GTA) in (A), and then we derive

$$\mathbb{P}_{(\mathbf{X},y)\sim\mathcal{D}} \left[ \min_{\|\boldsymbol{\xi}\|_2 \leq \delta} y f_{\mathbf{W}^{(T)}}(\mathbf{X} + \boldsymbol{\xi}) < 0 \right] \geq \mathbb{P}_{(\mathbf{X},y)\sim\mathcal{D}} \left[ y f_{\mathbf{W}^{(T)}}(\mathcal{A}(\mathbf{X})) < 0 \right]$$

$$= \mathbb{P}_{(\mathbf{X},y)\sim\mathcal{D}} \left[ \alpha^3 \sum_{r=1}^m \left( u_r^{(T)} \right)^3 (1-\gamma)^3 + y \sum_{r=1}^m \sum_{j \in [P]\backslash\text{signal}(\mathbf{X})} \left\langle \mathbf{w}_r^{(T)}, \mathbf{X}[j] \right\rangle^3 < 0 \right]$$

$$\geq \frac{1}{2} \mathbb{P}_{(\mathbf{X},y)\sim\mathcal{D}} \left[ \left| \sum_{r=1}^m \sum_{j \in [P]\backslash\text{signal}(\mathbf{X})} \left\langle \boldsymbol{\beta}_r^{(T)}, \mathbf{X}[j] \right\rangle^3 \right| \geq \tilde{\Omega} \left( (1-\gamma)^3 \right) \right] \geq \frac{1}{2} \left( 1 - \frac{\tilde{O}(d)}{2^d} \right) = \frac{1}{2} - o(1),$$

where we use Lemma A.4 in the last inequality. $\qquad\square$

# D  PROOF FOR SECTION 5

We prove Theorem 5.2 by using ReLU network to approximate $f_{\mathcal{S}}$ proposed in Section 1.

**Theorem D.1.** *(Restatement of Theorem 5.2) Under Assumption 5.1, with $N-$sample training dataset $\mathcal{S} = \{(\boldsymbol{X}_1, y_1), (\boldsymbol{X}_2, y_2), \dots, (\boldsymbol{X}_N, y_N)\}$ drawn from the data distribution $\mathcal{D}$, there exists a CGRO classifier that can be represented as a ReLU network with $\mathrm{poly}(D) + \tilde{O}(ND)$ parameters, which means that, under the distribution $\mathcal{D}$ and dataset $\mathcal{S}$, the network achieves zero clean test and robust training errors but its robust test error is at least $\Omega(1)$.*

*Proof.* First, we give the following useful results about function approximation by ReLU nets.

**Lemma D.2.** *(Yarotsky, 2017, Proposition 2) The function $f(x) = x^2$ on the segment $[0, 1]$ can be approximated with any error $\epsilon > 0$ by a ReLU network having the depth and the number of weights and computation units $O(\log(1/\epsilon))$.*

**Lemma D.3.** *(Yarotsky, 2017, Proposition 3) Let $\epsilon > 0$, $0 < a < b$ and $B \geq 1$ be given. There exists a function $\widetilde{\times} : [0, B]^2 \to [0, B^2]$ computed by a ReLU network with $O\left(\log^2\left(\epsilon^{-1}B\right)\right)$ parameters such that*

$$\sup_{x,y \in [0,B]} \left|\widetilde{\times}(x, y) - xy\right| \leq \epsilon,$$

*and $\widetilde{\times}(x, y) = 0$ if $xy = 0$.*

Since for $\forall \boldsymbol{X}_0 \in [0, 1]^D$, the $\ell_2-$distance function $\|\boldsymbol{X} - \boldsymbol{X}_0\|^2 = \sum_{i=1}^{D} |\boldsymbol{X}^{(i)} - \boldsymbol{X}_0^{(i)}|^2$, by using Lemma D.2, there exists a function $\phi_1$ computed by a ReLU network with $\mathcal{O}\left(D\log\left(\epsilon_1^{-1}D\right)\right)$ parameters such that $\sup_{\boldsymbol{X} \in [0,1]^D} \left|\phi_1(\boldsymbol{X}) - \|\boldsymbol{X} - \boldsymbol{X}_0\|^2\right| \leq \epsilon_1$.

Return to our main proof back, indeed, functions computed by ReLU networks are piecewise linear but the indicator functions are not continuous, so we need to relax the indicator such that $\hat{I}_{\mathrm{soft}}(x) = 1$ for $x \leq \delta + \epsilon_0$, $\hat{I}_{\mathrm{soft}}(x) = 0$ for $x \geq R - \delta\epsilon_0$ and $\hat{I}_{\mathrm{soft}}$ is linear in $(\delta + \epsilon_0, R - \delta\epsilon_0)$ by using only two ReLU neurons, where $\epsilon_0$ is sufficient small for approximation.

Now, we notice that the constructed function $f_{\mathcal{S}}$ can be re-written as

$$f_{\mathcal{S}}(\boldsymbol{X}) = f_{\mathrm{clean}}(\boldsymbol{X})\left(1 - \mathbb{I}\{\boldsymbol{X} \in \cup_{i=1}^{N}\mathbb{B}_2(\boldsymbol{X}_i, \delta)\}\right) + \sum_{i=1}^{N} y_i\mathbb{I}\{\boldsymbol{X} \in \mathbb{B}_2(\boldsymbol{X}_i, \delta)\}$$

$$= f_{\mathrm{clean}}(\boldsymbol{X}) + \sum_{i=1}^{N}(y_i - f_{\mathrm{clean}}(\boldsymbol{X}))\mathbb{I}\{\|\boldsymbol{X} - \boldsymbol{X_i}\|_2^2 \leq \delta^2\}.$$

Combined with Lemma D.2, Lemma D.3 and the relaxed indicator, we know that there exists a ReLU net $h$ with at most $\mathrm{poly}(D) + \tilde{O}(ND)$ parameters such that $|h - f_{\mathcal{S}}| = o(1)$ for all input $\boldsymbol{X} \in [0, 1]^D$. Thus, it is easy to check that $h$ belongs to CGRO classifiers.  $\square$

Next, we prove Theorem 5.3 by using the VC-dimension theory.

**Theorem D.4.** *(Restatement of Theorem 5.3) Let $\mathcal{F}_M$ be the family of function represented by ReLU networks with at most $M$ parameters. There exists a number $M_D = \Omega(\exp(D))$ and a distribution $\mathcal{D}$ satisfying Assumption 5.1 such that, for any classifier in the family $\mathcal{F}_{M_D}$, under the distribution $\mathcal{D}$, the robust test error is at least $\Omega(1)$.*

*Proof.* Now, we notice that ReLU networks are piece-wise linear functions. Montufar et al. (2014) study the number of local linear regions, which provides the following result.

**Proposition D.5.** *The maximal number of linear regions of the functions computed by any ReLU network with a total of $n$ hidden units is bounded from above by $2^n$.*

Thus, for a given clean classifier $f_{\text{clean}}$ represented by a ReLU net with $\text{poly}(D)$ parameters, we know there exists at least a local region $V$ such that decision boundary of $f_{\text{clean}}$ is linear hyperplane in $V$. And we assume that the hyperplane is $\boldsymbol{X}^{(D)} = \frac{1}{2}$.

Then, let $V'$ be the projection of $V$ on the decision boundary of $f_{\text{clean}}$, and $\mathcal{P}$ be an $2\delta$-packing of $V'$. Since the packing number $\mathcal{P}(V', \|\cdot\|, 2\delta) \geq \mathcal{C}(V', \|\cdot\|_2, 2\delta) = \exp(\Omega(D))$, where $\mathcal{C}(\Theta, \|\cdot\|, \delta)$ is the $\delta$-covering number of a set $\Theta$. For any $\epsilon_0 \in (0, 1)$, we can consider the construction

$$S_\phi = \left\{ \left(\boldsymbol{x}, \frac{1}{2} + \epsilon_0 \cdot \phi(\boldsymbol{x})\right) : \boldsymbol{x} \in \mathcal{P} \right\},$$

where $\phi : \mathcal{P} \to \{-1, +1\}$ is an arbitrary mapping. It's easy to see that all points in $S_\phi$ with first $D - 1$ components satisfying $\|\boldsymbol{x}\|_2 \leq \sqrt{1 - \epsilon_0^2}$ are in $V'$, so that by choosing $\epsilon_0$ sufficiently small, we can guarantee that $|S_\phi \cap V| = \exp(\Omega(D))$. For convenience we just replace $S_\phi$ with $S_\phi \cap V$ from now on.

Let $A_\phi = S_\phi \cap \left\{ \boldsymbol{X} \in V : \boldsymbol{x}^{(D)} > \frac{1}{2} \right\}$, $B_\phi = S_\phi - A_\phi$. It's easy to see that for arbitrary $\phi$, the construction is linear-separable and satisfies $2\delta$-separability.

Assume that for any choices of $\phi$, the induced sets $A_\phi$ and $B_\phi$ can always be robustly classified with $(O(\delta), 1 - \mu)$-accuracy by a ReLU network with at most $M$ parameters. Then, we can construct an *enveloping network* $F_\theta$ with $M - 1$ hidden layers, $M$ neurons per layer and at most $M^3$ parameters such that any network with size $\leq M$ can be embedded into this envelope network. As a result, $F_\theta$ is capable of $(O(\delta), 1 - \mu)$-robustly classify any sets $A_\phi, B_\phi$ induced by arbitrary choices of $\phi$. We use $R_\phi$ to denote the subset of $S_\phi = A_\phi \cup B_\phi$ satisfying $|R_\phi| = (1 - \mu)|S_\phi| = \exp(\Omega(D))$ such that $R_\phi$ can be $O(\delta)$-robustly classified.

Next, we estimate the lower and upper bounds for the cardinal number of the vector set

$$R := \{(f(\boldsymbol{x}))_{\boldsymbol{x} \in \mathcal{P}} | f \in \mathcal{F}_{M_D}\}.$$

Let $n$ denote $|\mathcal{P}|$, then we have

$$R = \{(f(\boldsymbol{x}_1), f(\boldsymbol{x}_2), ... f(\boldsymbol{x}_n)) | f \in \mathcal{F}_{M_D}\},$$

where $\mathcal{P} = \{\boldsymbol{x}_1, \boldsymbol{x}_2, ..., \boldsymbol{x}_n\}$.

On one hand, we know that for any $u \in \{-1, 1\}^n$, there exists a $v \in R$ such that $d_H(u, v) \leq \alpha n$, where $d_H(\cdot, \cdot)$ denotes the Hamming distance, then we have

$$|R| \geq \mathcal{N}(\{-1, 1\}^n, d_H, \mu n) \geq \frac{2^n}{\sum_{i=0}^{\mu n} \binom{n}{i}}.$$

On the other hand, by applying Lemma A.8, we have

$$\frac{2^n}{\sum_{i=1}^{\mu n} \binom{n}{i}} \leq |R| \leq \Pi_{\mathcal{F}_{M_D}}(n) \leq \sum_{j=0}^{l} \binom{n}{j}.$$

where $l$ is the VC-dimension of $\mathcal{F}_{M_D}$. In fact, we can derive $l = \Omega(n)$ when $\mu$ is a small constant. Assume that $l < n - 1$, then we have $\sum_{j=0}^{l} \binom{n}{j} \leq (en/l)^l$ and $\sum_{i=1}^{\mu n} \binom{n}{i} \leq (e/\mu)^{\mu n}$, so

$$\frac{2^n}{(e/\mu)^{\mu n}} \leq |R| \leq (en/l)^l.$$

We define a function $h(x)$ as $h(x) = (e/x)^x$, then we derive

$$2 \leq \left(\frac{e}{\mu}\right)^\mu \left(\frac{e}{l/n}\right)^{l/n} = h(\mu)h(l/n).$$

When $\mu$ is sufficient small, $l/n \geq C(\mu)$ that is a constant only depending on $\mu$, which implies $l = \Omega(n)$. Finally, by using Lemma A.7 and $n = |\mathcal{P}| = \exp(\Omega(D))$, we know $M_D = \exp(\Omega(D))$. $\quad\square$

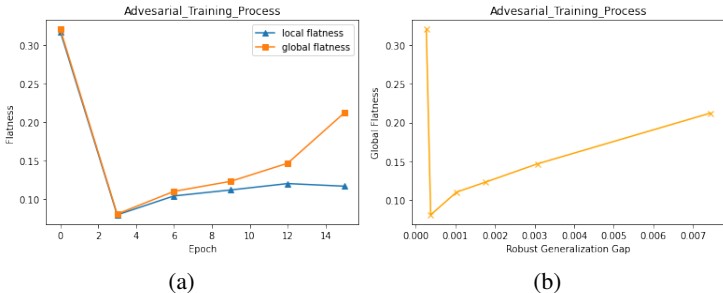

(a)                                   (b)

Figure 3: **Left:** Local and Global Flatness During Adversarial Training on CIFAR10; **Right:** The Relation Between Robust Generalization Gap and Global Flatness on CIFAR10.

# E   ROBUST GENERALIZATION BOUND BASED ON GLOBAL FLATNESS

In this section, we prove a novel robust generalization bound that mainly depends on global flatness of loss landscape. We consider $\ell_p-$adversarial robustness with perturbation radius $\delta$ and we use $\mathcal{L}_{\text{clean}}$, $\mathcal{L}_{\text{adv}}(f)$ and $\widehat{\mathcal{L}}_{\text{adv}}(f)$ to denote the clean test risk, the adversarial test risk and the adversarial empirical risk w.r.t. the model $f$, respectively. We also assume $\frac{1}{p} + \frac{1}{q} = 1$ for the next results.

**Theorem E.1.** *(Robust Generalization Bound) Let $\mathcal{D}$ be the underlying distribution with a smooth density function, and $N-$sample training dataset $\mathcal{S} = \{(\boldsymbol{X}_1, y_1), (\boldsymbol{X}_2, y_2), \ldots, (\boldsymbol{X}_N, y_N)\}$ is i.i.d. drawn from $\mathcal{D}$. Then, with high probability, it holds that,*

$$\mathcal{L}_{adv}(f) \leq \widehat{\mathcal{L}}_{adv}(f) + N^{-\frac{1}{D+2}} O\left(\underbrace{\mathbb{E}_{(\boldsymbol{X},y)\sim\mathcal{D}}\left[\max_{\|\boldsymbol{\xi}\|_p \leq \delta} \|\nabla_{\boldsymbol{X}}\mathcal{L}(f(\boldsymbol{X}+\boldsymbol{\xi}), y)\|_q\right]}_{global\ flatness}\right).$$

This generalization bound shows that robust generalization gap can be dominated by global flatness of loss landscape. And we also have the lower bound of robust generalization gap stated as follow.

**Proposition E.2.** *Let $\mathcal{D}$ be the underlying distribution with a smooth density function, then we have*

$$\mathcal{L}_{adv}(f) - \mathcal{L}_{clean}(f) = \Omega\left(\delta\mathbb{E}_{(\boldsymbol{X},y)\sim\mathcal{D}}\left[\|\nabla_{\boldsymbol{X}}\mathcal{L}(f(\boldsymbol{X}), y)\|_q\right]\right).$$

Theorem E.1 and Proposition E.2 manifest that robust generalization gap is very related to global flatness. However, although adversarial training achieves good local flatness by robust memorization on training data, the model lacks global flatness, which leads to robust overfitting.

This point is also verified by numerical experiment on CIFAR10 (see results in Figure 3). First, global flatness grows much faster than local flatness in practice. Second, with global flatness increasing during training process, it causes an increase of robust generalization gap.

## F  PROOF FOR SECTION E

**Theorem F.1.** *(Restatement of Theorem E.1) Let $\mathcal{D}$ be the underlying distribution with a smooth density function, and $N-$sample training dataset $\mathcal{S} = \{(\boldsymbol{X}_1, y_1), (\boldsymbol{X}_2, y_2), \ldots, (\boldsymbol{X}_N, y_N)\}$ is i.i.d. drawn from $\mathcal{D}$. Then, with high probability, it holds that,*

$$\mathcal{L}_{adv}(f) \leq \widehat{\mathcal{L}}_{adv}(f) + N^{-\frac{1}{D+2}} O\left(\underbrace{\mathbb{E}_{(\boldsymbol{X},y)\sim\mathcal{D}}\left[\max_{\|\boldsymbol{\xi}\|_p \leq \delta} \|\nabla_{\boldsymbol{X}} \mathcal{L}(f(\boldsymbol{X}+\boldsymbol{\xi}), y)\|_q\right]}_{\textit{global flatness}}\right).$$

*Proof.* Indeed, we notice the following loss decomposition,

$$\mathcal{L}_{\text{adv}}(f) - \widehat{\mathcal{L}}_{\text{adv}}(f) = \left(\mathcal{L}_{\text{clean}}(f) - \widehat{\mathcal{L}}_{\text{adv}}(f)\right) + \left(\mathcal{L}_{\text{adv}}(f) - \mathcal{L}_{\text{clean}}(f)\right).$$

To bound the first term, by applying $\lambda_i$ to denote kernel density estimation (KDE) proposed in Petzka et al. (2020), then we derive

$$\mathcal{L}_{\text{clean}}(f) - \widehat{\mathcal{L}}_{\text{adv}}(f) = \mathbb{E}_{(\boldsymbol{X},y)\sim\mathcal{D}}[\mathcal{L}(f(\boldsymbol{X}), y)] - \frac{1}{N}\sum_{i=1}^{N}\max_{\|\boldsymbol{\xi}\|_p \leq \delta}\mathcal{L}(f(\boldsymbol{X}_i + \boldsymbol{\xi}, y_i))$$

$$\leq \mathbb{E}_{(\boldsymbol{X},y)\sim\mathcal{D}}[\mathcal{L}(f(\boldsymbol{X}), y)] - \frac{1}{N}\sum_{i=1}^{N}\mathbb{E}_{\boldsymbol{\xi}\sim\lambda_i}[\mathcal{L}(f(\boldsymbol{X}_i + \boldsymbol{\xi}), y_i)]$$

$$= \int_{\boldsymbol{X}} p_{\mathcal{D}}(\boldsymbol{X})\mathcal{L}(f(\boldsymbol{X}), y)d\boldsymbol{X} - \int_{\boldsymbol{X}} p_{\mathcal{S}}(\boldsymbol{X})\mathcal{L}(f(\boldsymbol{X}), y)d\boldsymbol{X}$$

$$\leq \underbrace{\left|\int_{\boldsymbol{X}} (p_{\mathcal{D}}(\boldsymbol{X}) - \mathbb{E}_{\mathcal{S}}[p_{\mathcal{S}}(\boldsymbol{X})])\mathcal{L}(f(\boldsymbol{X}), y(\boldsymbol{X}))d\boldsymbol{X}\right|}_{(I)}$$

$$+ \underbrace{\left|\int_{\boldsymbol{X}} (\mathbb{E}_{\mathcal{S}}[p_{\mathcal{S}}(\boldsymbol{X})] - p_{\mathcal{S}}(\boldsymbol{X}))\mathcal{L}(f(\boldsymbol{X}), y(\boldsymbol{X}))d\boldsymbol{X}\right|}_{(II)},$$

where $p_{\mathcal{D}}(\boldsymbol{X})$ is the density function of the distribution $\mathcal{D}$, and $p_{\mathcal{S}}(\boldsymbol{X})$ is the KDE of point $\boldsymbol{X}$.

With the smoothness of density function of $\mathcal{D}$ and Silverman (2018), we know that $(I) = O(\delta^2)$.

For (II), by using Chebychef inequality and Silverman (2018), with probability $1 - \Delta$, we have

$$(II) = O(\Delta^{-\frac{1}{2}} N^{-\frac{1}{2}} \delta^{-\frac{D}{2}} + N^{-2}).$$

On the other hand, by Taylor expansion, we know

$$\mathcal{L}_{\text{adv}}(f) - \mathcal{L}_{\text{clean}}(f) \leq O(\delta)\mathbb{E}_{(\boldsymbol{X},y)\sim\mathcal{D}}\left[\max_{\|\boldsymbol{\xi}\|_p \leq \delta} \|\nabla_{\boldsymbol{X}} \mathcal{L}(f(\boldsymbol{X}+\boldsymbol{\xi}), y)\|_q\right].$$

Combined with the bounds for $(I)$ and $(II)$, we can derive Theorem E.1. $\qquad\square$

