# OpenReview forum: "Why Clean Generalization and Robust Overfitting Both Happen in Adversarial Training"
_ICLR.cc/2024/Conference — Submitted to ICLR 2024_

### Official Review · Reviewer_LZqy · 2023-10-27

**Soundness:** 2 fair
**Presentation:** 3 good
**Contribution:** 2 fair
**Rating:** 3
**Confidence:** 3

**Summary:**

This paper provides a theoretical understanding to the empirical finding of clean generalization and  robust overfitting (CGRO). They investigate this problem through feature learning theory. They mainly focus on patch-structured dataset and one-hidden-layer CNN model. The results suggest that the network can partially learn the true feature but exactly memorize the spurious features from  adversarial examples which leads to the CGRO phenomenon. They also provide an analysis of the efficiency of classifier using representation complexity theory.

**Strengths:**

* The paper is well written overall.
* The investigated phenomenon is interesting and worth exploring.
* The authors reveal a three-stage training process for adversarial training dynamics, which may offer insights to the community.

**Weaknesses:**

The authors might have made enough effort to complete this work; however, I have the following concerns about this work:
From a theoretical perspective:
* The assumption of the patch data is somewhat weird to me. The realistic images are much more complicated and cannot be simplified as a patch structual. Specifically, the authors assume $\alpha=d^{0.249}$ and $\sigma^{-0.509}$. I am puzzled as to why they chose specific values of $0.249$ and $-0.509$? Does the real data distribution satisfy this assumption?

From an experimental perspective:
* More experiments are needed to prove the validity of the theory. For example, the authors only chose a ResNet9 architecture for CIFAR10. In my opinion, they should validate the theory based on larger models that are widely used in the adversarial training field, such as ResNet-50.
* It would be better to empirically certify that the data assumption and the parameter settings are reasonable.

**Questions:**

Please refer to the Weaknesses.

---

> ### Author Response · Authors · 2023-11-22
> **Response to Reviewer LZqy:**
>
> We thank the reviewer of the insightful feedback. We address the reviewer’s concerns as below, and we are happy to clarify any follow-up questions/confusions.
>
> **Question**  The assumption of the patch data is somewhat weird to me. The realistic images are much more complicated and cannot be simplified as a patch structual. Specifically, the authors assume $\alpha=d^{0.249}$ and $\sigma^{-0.509}$. I am puzzled as to why they chose specific values of 0.249 and -0.509 ? Does the real data distribution satisfy this assumption?
>
> **Answer** Indeed, under our theoretical framework, the meaningful signal patch has the form $X[signal(X)] = \alpha y w^{\*}$ and the noise patch has the form $X[j] \sim N(0,(I_d-w^{\*}w^{\*T})\sigma^2)$, so the norm of meaningful signal patch is \alpha and the norm of noise patch w.h.p. is $\Theta(\sigma\sqrt{d-1})$, which enables meaningful signal is stronger than noise when we set $\alpha = d^{0.249}polylog(d)$, $\sigma = d^{-0.509}$. And if the data cannot satisfy this assumption, in other word, if the signal-to-noise ratio is small, then there even exists no clean generalization, which has been theoretically shown in the previous work [1][2][3] under the similar patch-structured setting. In this paper, we aim to provide a theoretical understanding of CGRO phenomenon in adversarial training, thereby the assumption that meaningful signal is stronger than random noise is reasonable and necessary. We will add some discussion to clarify this point about the signal and noise in the revision of paper. Thanks for the reviewer’s suggestion.
>
> [1] Towards understanding how momentum improves generalization in deep learning.
>
>
> [2] Benign Overfitting in Two-layer Convolutional Neural Networks.
>
>
> [3] Benign Overfitting in Two-layer ReLU Convolutional Neural Networks.

---

### Official Review · Reviewer_EEZ9 · 2023-10-27

**Soundness:** 3 good
**Presentation:** 3 good
**Contribution:** 3 good
**Rating:** 6
**Confidence:** 3

**Summary:**

This paper use one specific data distribution to show that the adversarial training suffers from robust over-fitting problem while the clean testing performance is good.

**Strengths:**

The overall quality of this paper is good. The logic is clear, and the observation in the theory is interesting. Although many existing literature tries to explain the robust over-fitting problem, this work is the first one of its type and provides very useful insights. The writing is also clear to understand.

**Weaknesses:**

My only concern of this paper is the choice of the activation function. This paper uses a cubic activation function, which is not commonly used in practice. In existing literature, e.g., Allen-Zhu et al. (2022), they consider two-layer MLP with ReLU activation function. The authors of this submission need to provide some illustration on why to use the cubic activation function. More specifically, if we replace the cubic activation function with ReLU or Sigmoid function, which part of the proof will face difficulty? I read through the proof and got a picture of the proof, but could not identify the exact place where the cubic activation function plays a crucial rule.

Another concern is that this paper provides experiments in MNIST and CIFAR-10, but without any simulation study to verify the correctness of the theorem. The authors are encouraged to use simulation (follow the exact assumption in the theory) to at least verify and visualize the observations in the theorem.

Zeyuan Allen-Zhu and Yuanzhi Li. Feature purification: How adversarial training performs robust deep learning. In 2021 IEEE 62nd Annual Symposium on Foundations of Computer Science (FOCS), pp. 977–988. IEEE, 2022.

**Questions:**

Please address my two concerns above.

---

> ### Author Response · Authors · 2023-11-22
> **Response to Reviewer EEZ9:**
>
> We thank the reviewer of the insightful feedback and improvement suggestions. We would respectively address the major concerns below, and we are happy to discuss any follow-up questions.
>
> **Question** My only concern of this paper is the choice of the activation function. This paper uses a cubic activation function, which is not commonly used in practice. In existing literature
>
> **Answer**  Different from some previous work analyzing the behavior of linear classifier in adversarial training, we improve the expressive power of model by using non-linear cubic activation that can characterize the data structure, in order that we can analyze learning process more precisely. Moreover, under our theoretical framework, the results on feature learning process still hold for more general activation function $z^{q}$, where q is arbitrary odd number at least three to balance the positive-labeled data and negative-labeled data.

---

> > ### Comment · Reviewer_EEZ9 · 2023-11-22
> >
> > Thanks for your comment. Could you explicitly specify which equation/inequality exactly uses any property of $z^q$ so that quadratic activation will fail?

---

> > > ### Author Response · Authors · 2023-11-22
> > > **Response to Reviewer EEZ9:**
> > >
> > > Thank the reviewer for the reply. Indeed, if we use quadratic activation function, the function value $f(X; W)=\sum_{r=1}^{m}\sum_{j=1}^{P}\langle w_r, X[j]\rangle^{2} \geq 0$ for any time, which makes it is impossible for the classifier to correctly classify the data with negative label $-1$. However, with observed clean generalization, the trained network is able to classify both positive and negative classes in practice. Thus, we use odd-deg activation function $z^{q}$ to address this problem.

---

### Official Review · Reviewer_6rXn · 2023-11-01

**Soundness:** 2 fair
**Presentation:** 3 good
**Contribution:** 3 good
**Rating:** 5
**Confidence:** 2

**Summary:**

This paper offers theoretical intuitions about why clean generalization and robust overfitting happen simultaneously in robust neural classifiers. Specifically, the results consist of two parts. The first part shows that for a one-hidden-layer cubic-activated convolutional neural network trained on a patched dataset with a transfer attack, the training behavior follows a three-phase evolution, demonstrating that in later phases of neural network training, the neural network remembers spurious features. The second part shows that for well-separated and neural-separable datasets, a CGRO classifier with $poly(D) + \tilde{O}(ND)$ parameters exists, and this parameter count is much less than potentially required by a truly robust classifier.

**Strengths:**

This paper proves some very cool results that correspond to real-world observations of adversarial training dynamics. Section 5 provides solid intuitions to explain the phenomenon of practical robust classifiers requiring much more parameters than their non-robust counterparts, even with relatively small datasets. The paper is well-structured, and the story is relatively complete.

**Weaknesses:**

- The majority of results in this paper consider the binary classification setting. This should be made more clear in the abstract/introduction.
- The assumptions made in Sections 3 and 4 seem arbitrary and unreasonably strong. In particular, an uncommon cubic activation function is chosen, and the correlation metrics are defined to be cubic without much explanation. Furthermore, a non-standard transfer attack that follows an atypical gradient update rule (Eq. (A)) is used for adversarial training in Section 3.3. On the other hand, the authors managed to prove some meaningful results using these strong assumptions, so this is a tough call. On the other hand, the results in Section 5 are much more general and practical, in my opinion.
- The notations section reads, "We use $poly(\cdot)$ to denote the polynomial order, and $polylog(\cdot)$ to denote the some polynomial in log order." This sentence is quite confusing. If I understood it correctly, a much better sentence would be, "We use $poly(\cdot)$ to denote a polynomial function, and use $polylog(\cdot)$ to denote the composition of a polynomial function over a log function."
- The term "$poly (\cdot)$" itself is quite ambiguous. It would be clearer to mention what polynomial it is explicitly.
- Since I found the results in Section 5 to be more practical and thus potentially more important, I would recommend adding a comparison in the experiments regarding different model sizes. It would be very supportive if you can show that when the model becomes larger, first the robust training loss decreases but the robust generalization gap remains large, and then when the model gets even larger, the robust generalization gap gradually decreases.

**Questions:**

- For the patch dataset, why is $\alpha$ assumed to be proportional to $d^{0.249}$? Where is the number $0.249$ from? Similarly, where is the number $-0.509$ from in the assumption about $\sigma$?
- The sentence above Section 4 reads, "Intuitively, when the model $f_W$ has achieve (typo here, should be achieved) mid-high clean test accuracy, the decision boundary of W will have a significant correlation with the separating plane of $g$, which thus makes adversarial examples generated by $g$ transferable." Is there any justifications or source for this statement?
- The first sentence of Section 4.1 reads, "which is widely applied in theoretical works explore what and how neural networks learn in different tasks." Check grammar.
- In Figures 2c and 2d, the gradient magnitude and the value change on the CIFAR-10 dataset seems to increase at a radius smaller than $\frac{8}{255}$. It would also make more sense to include a training data example and a test data example.

---

> ### Author Response · Authors · 2023-11-22
> **Response to Reviewer 6rXn:**
>
> We thank the reviewer of the insightful feedback. We address the reviewer’s concerns as below, and we are happy to clarify any follow-up questions/confusions.
>
> **Q1** For the patch dataset, why is $\alpha$ assumed to be proportional to $d^{0.249}$ ? Where is the number 0.249 from? Similarly, where is the number -0.509 from in the assumption about $\sigma$ ?
>
> **A1** Indeed, under our theoretical framework, the meaningful signal patch has the form $X[signal(X)] = \alpha y w^{\*}$ and the noise patch has the form $X[j] \sim N(0,(I_d-w^{\*}w^{\*T})\sigma^2)$, so the norm of meaningful signal patch is \alpha and the norm of noise patch w.h.p. is $\Theta(\sigma\sqrt{d-1})$, which enables meaningful signal is stronger than noise when we set $\alpha = d^{0.249}polylog(d)$, $\sigma = d^{-0.509}$. And if the data cannot satisfy this assumption, in other word, if the signal-to-noise ratio is small, then there even exists no clean generalization, which has been theoretically shown in the previous work [1][2][3] under the similar patch-structured setting. In this paper, we aim to provide a theoretical understanding of CGRO phenomenon in adversarial training, thereby the assumption that meaningful signal is stronger than random noise is reasonable and necessary. We will add some discussion to clarify this point about the signal and noise in the revision of paper. Thanks for the reviewer’s suggestion.
>
> **Q2** The sentence above Section 4 reads, "Intuitively, when the model $f_W$ has achieve (typo here, should be achieved) mid-high clean test accuracy, the decision boundary of W will have a significant correlation with the separating plane of $g$, which thus makes adversarial examples generated by $g$ transferable." Is there any justifications or source for this statement?
>
> **A2** Theoretically, traditional formulation of adversarial training algorithms is a highly non-convex and non-concave min-max optimization, under which, in general, analyzing global convergence is NP-hard problem [4]. To overcome this hardness barrier, we leverage geometry-inspired attack as adversarial training method, under which we propose a non-trivial three-stage analysis technique to decouple the complicated feature learning process as follows.
>
> &emsp;**Phase I**: At the beginning, the signal component of lottery tickets winner $max_{r\in [m]}u_{r}^{(t)}$ increases quadratically (Lemma 4.2). At this point, the model starts to learn partial true feature.
>
> &emsp;**Phase II**: Once the maximum signal component $max_{r\in [m]}u_{r}^{(t)}$ attains the order $\tilde{\Omega}(\alpha^{-1})$, the growth of signal component nearly stop updating since that the increment of signal component is now mostly dominated by the noise component (Lemma 4.3).
>
> &emsp;**Phase III**: After that, by the quadratic increment of noise component (Lemma 4.4), the total noise $V_{i}^{(t)}$ eventually attains the order $\Omega(1)$, which implies the model memorizes the spurious feature (data-wise noise) in final.
>
> &ensp;**On the empirical side**, it is verified that it has a comparable performance with classical adversarial training methods such as FGSM and PGD attacks (e.g., [5][6]). Understand the feature learning process of FGSM and PGD attacks is an existing future direction.
>
>
>
> [1] Towards understanding how momentum improves generalization in deep learning.
> [2] Benign Overfitting in Two-layer Convolutional Neural Networks.
> [3] Benign Overfitting in Two-layer ReLU Convolutional Neural Networks.
>
>
> [4] Some NP-complete problems in quadratic and nonlinear programming.
>
> [5] Deepfool: a simple and accurate method to fool deep neural networks.
>
> [6] Geometry-inspired top-k adversarial perturbations.
>
> **Q3** The first sentence of Section 4.1 reads, "which is widely applied in theoretical works explore what and how neural networks learn in different tasks." Check grammar.
>
> **A3** this sentence should be “which is widely applied in theoretical works that study what and how neural networks learn in different tasks”.

---

### Official Review · Reviewer_rmWV · 2023-11-02

**Soundness:** 3 good
**Presentation:** 1 poor
**Contribution:** 2 fair
**Rating:** 5
**Confidence:** 3

**Summary:**

Machine learning models that are subject to adversarial attacks perform poorly out-of-sample when these attacks are not incorporated into the model during the training phase. Adversarial training, minimizing the empirical error while imitating the attacks during the training phase, has thus emerged as the *de facto* standard for training models to hedge against adversarial attacks. It has been increasingly observed that adversarially robust models still suffer from overfitting (called "robust overfitting"), even though the training set was perturbed via adversarial attacks. On the contrary, adversarially trained models generalize fairly well when the test set remains "clean", that is when there is no adversarial attack on the test set the model "clean generalizes". In this work, specifically for CNNs, the authors analyze why a model can simultaneously robust overfit and clean generalize. By using techniques from feature learning theory, they show that along the training of CNN, the network learns the "clean" relationship between the features and the classes, while memorizing the adversarially perturbed training instances.

**Strengths:**

The paper follows up on modern research, cites very relevant resources, and is mathematically sound. The authors do a great job on the balance between asymptotic and exact notation/results. I also like the proof technique which is summarized in Section 4.3., which is original to this work.

**Weaknesses:**

Although the underlying proofs of this paper are interesting, I think the work is not ready for acceptance. Before I summarize my concerns I would like to note that I am familiar with the adversarial training and robust optimization literature, but I am not an expert in feature learning or CNNs. Therefore, I will have a very open mind and stay active during the discussion period to read the authors' responses.
-- --
**Major Concern 1: Robust Optimization overlooked**

I find it hard to agree with the motivation of this paper, especially for the "clean generalization" part. If we train a model adversarially but there is no attack in the test phase, this is precisely traditional robust optimization (not adversarial or distributionally robust, I mean the traditional version). We can think of the data at hand as noisy, that is if the observed feature values $X$ are noisy observations of the true feature values, then we build uncertainty sets around them via the balls $\mathbb{B}_p(X_i, \delta)$ -- see, for example, "Robust classification" by Bertsimas et al. (2019). In robust optimization, the "test phase" does not have any noise as the historical data is noisy but the risk is with respect to the true data. Moreover, the earliest papers already established the "clean generalization" result -- optimizing the worst-case during training does not give solutions that are significantly away from the nominal solutions (e.g., "Robust solutions of Linear Programming problems contaminated with uncertain data" by Ben-Tal and Nemirovski, 2000). On the contrary, robust optimization can even improve the out-of-sample performance of ML models (see, e.g., "Regularization via mass transportation" by Shafieezadeh-Abadeh et al. (2019) for the reasoning for the specific case of Wasserstein distributionally robust optimization and "Certified Robust Neural Networks: Generalization and Corruption Resistance" by Bennouna et al. (2023) for the connection between robust optimization and adversarial training).

As a reader, I would have preferred to have further discussion on this.
-- --
**Major Concern 2: The paper is not written well, and includes several typos and errors**

In the "Questions" section I will list some examples, but the paper is not ready yet in my view. I had to read many times before I understood most of the claims.
-- --
**Major Concern 3: The setting is very specific, and conclusions are general**

The paper, from its beginning, introduces assumptions, parameterizations, architecture, and dataset structures without really discussing why. Overall, the definitions are ad-hoc and there is an imposed assumption that the reader will have a deep understanding of all three: (i) adversarial robustness, (ii) feature learning theory, (iii) CNNs. I could not even get a gist of the ideas related to the literature that I am not used to as there is no introduction to: signals, patches, spurious feature learning, etc. I will add more in the "Questions" section. The main message of this paper is around robust overfitting (and how it can clean generalize, for example), and this image-data-related work is just a special case, many readers from the adversarial training field will be very interested in understanding the reason behind the phenomenon that is being studied here -- so I would strongly recommend a clearer and more inclusive language.

**Questions:**

Note that some questions are rhetorical and would be great to fix them in the paper.
- "clean data" -> could the authors please specify this is attack-free or not-perturbed data when "clean data" is first mentioned?
- Abstract and later parts of the paper mentions "more general data assumption" -> this is vague
- Introduction: "including in" -> what does this mean (WDM?)
- "can make well-trained classifiers confused" -> WDM?
- "robust generalization requires more data" -> compared to what?
- "well-separated, which means half of the distance between the positive data and negative data is larger than ..." -> I cannot follow.
- (C): what if the second summation's indicator is $=1$ for more than one $i$?
- (C): Before (C) it is said "we consider the following function as" - > as what? That said, there is a period at the end of the line of (C) but the sentence continues.
- (C): There is a cyclical argument here because before (C) it is said "we **consider** the following function" but after the definition there is "inspired by the intuition" -> how can we get inspired from something that we just defined holistically?
- "Theorefore" -> typo
- The first contribution at the bottom of page 2: This is not a contribution, but a setting that you are working in to simplify the proofs.
- Second contribution: increases "quadratically" -> Quadratic in what? (Same for "polynomial" later on.) This does not have a meaning. Similarly, "partial true feature" does not tell much before one understands the later sections.
- Notations: $O(\cdot)$ before $\Omega(\cdot)$ -> is this a typo?
- Section 3.1: completely unclear. What is $X[\mathrm{signal}(X)]$? This is never defined? Even a "signal" is not mathematically defined. I also do not follow "meaningful signal" terminology. "norm of data" -> which norm, what data?
- Section 3.2, "we use a one-hidden layer CNN" -> why? There is no reasoning currently.
- Section 3.2, "we apply the cubic activation function" -> similarly, why? "as polynomial activations are standard" is not a convincing argument.
- Section 3.2: "linear model can achieve" and "which but fails to explain" have typos.
-  Section 3.3: "randomly sampled from the data distribution $\mathcal{D}$ -> independently?
- Question: Why is $\lambda$ not fixed as $1$ in (F)? Which papers keep it as a generic $\lambda > 0$?
- Question: Can the authors please discuss why $\delta = \alpha(1-  1/(\sqrt{d} \mathrm{polylog}(d)))$ is assumed? Why is the attack strength now "free"?
- Similarly, "we assume $\lambda \in [1 / \mathrm{poly}(d), 1)$" needs some convincing arguments.
- (O): Isn't $\lambda = 1$ in the adversarial training literature?
- "target classifier $g$" -> what is a target classifier?
- "to solve the minimization problem (O)" -> (O) is a function not a minimization problem.
- "when the model $f_W$ has achieve mid-high clean..." -> typo. "explore what and how neural networks" -> typo. "the model correctly classify the data" -> typo.
- "Spurious feature learning" -> Can the authors please carefully define "spurious"? It becomes clear from the proofs why we need this, but in the main paper it was hard to follow for me.
- Theorem 4.1: (1)(1) is a typo. Additionally, would it be clearer if "with high probability" was more formally described via confidence parameters?
- That said, I would like to note that I like the discussion before Section 4.2 and the explanation after Theorem 4.1.

---

> ### Author Response · Authors · 2023-11-22
> **Response to Reviewer rmWV:**
>
> We thank the reviewer of the insightful feedback and improvement suggestions. We would respectively address the major concerns below, and we are happy to discuss any follow-up questions.
>
> **Q1** Major Concern 1: Robust Optimization overlooked
> I find it hard to agree with the motivation of this paper, especially for the "clean generalization" part.
>
> **A1** while some works (Tsipras et al., 2018; Zhang et al., 2019; Hassani & Javanmard, 2022) point out that achieving robustness may hurt clean test accuracy, in most of the cases, it is observed that drop of robust test accuracy is much higher than drop of clean test accuracy in adversarial training (Madry et al., 2017; Schmidt et al., 2018; Raghunathan et al., 2019) (see in Figure 1, where clean test accuracy is more than 80% but robust test accuracy only attains nearly 50%). Namely, a weak version of benign overfitting (Zhang et al., 2017), which means that overparameterized deep neural networks can both fit random data powerfully and generalize well for unseen clean data, remains after adversarial training. Therefore, it is natural to ask the following question: What is the underlying mechanism that results in both Clean Generalization and Robust Overfitting (CGRO) during adversarial training? In this paper, We provide a theoretical understanding why clean generalization and robust overfitting both happen in adversarial training.
>
> **Q2** Major Concern 2: The paper is not written well, and includes several typos and errors
>
> **A2** Thank the reviewer for pointing out the typos in this paper, we will fix these in the revision of our paper.
>
> **Q3** Major Concern 3: The setting is very specific, and conclusions are general.
>
> **A3** Theoretically, traditional formulation of adversarial training algorithms is a highly non-convex and non-concave min-max optimization, under which, in general, analyzing global convergence is NP-hard problem [1]. To overcome this hardness barrier, we leverage geometry-inspired attack as adversarial training method, under which we propose a non-trivial three-stage analysis technique to decouple the complicated feature learning process as follows.
>
> &emsp;**Phase I**: At the beginning, the signal component of lottery tickets winner $max_{r\in [m]}u_{r}^{(t)}$ increases quadratically (Lemma 4.2). At this point, the model starts to learn partial true feature.
>
> &emsp;**Phase II**: Once the maximum signal component $max_{r\in [m]}u_{r}^{(t)}$ attains the order $\tilde{\Omega}(\alpha^{-1})$, the growth of signal component nearly stop updating since that the increment of signal component is now mostly dominated by the noise component (Lemma 4.3).
>
> &emsp;**Phase III**: After that, by the quadratic increment of noise component (Lemma 4.4), the total noise $V_{i}^{(t)}$ eventually attains the order $\Omega(1)$, which implies the model memorizes the spurious feature (data-wise noise) in final.
>
> &ensp;**On the empirical side**, it is verified that it has a comparable performance with classical adversarial training methods such as FGSM and PGD attacks (e.g., [2][3]). Understand the feature learning process of FGSM and PGD attacks is an existing future direction.
>
> In all, we present a theoretical understanding of clean generalization and robust overfitting (CGRO) phenomenon in adversarial training. Our main contribution is that, under our theoretical framework, we prove that neural network trained by adversarial training partially learns the true feature but memorizes the random noise in training data, which leads to CGRO phenomenon. In all, we believe that our work provides some theoretical insights into existing adversarial training methods. An important future work is to generalize our analysis of feature learning process to deep CNN models with other adversarial-example generative algorithms, such as FGSM and PGD attacks.
>
> [1] Some NP-complete problems in quadratic and nonlinear programming.
>
> [2] Deepfool: a simple and accurate method to fool deep neural networks.
>
> [3] Geometry-inspired top-k adversarial perturbations.

---

### Official Review · Reviewer_CpMF · 2023-11-04

**Soundness:** 3 good
**Presentation:** 3 good
**Contribution:** 2 fair
**Rating:** 5
**Confidence:** 4

**Summary:**

The paper studies the overfitting phenomenon in the context of adversarial training and attempts to understand how clean generalization and robust overfitting could happen at the same time.  The theoretical framework in section 3 considers a two-layer neural network with a cubic activation function. The adversarial loss is defined in (F)  and the analysis focuses on a first-order attack scheme defined in (A). Theorem 4.1 is the paper's main result indicating that the adversarial training algorithm will partially learn the true features while memorizing the spurious features. Some numerical results are discussed in section 6 to show local flatness around training data while the sharpness of the loss function outside the training radius.

**Strengths:**

1- The paper studies an interesting problem on the generalization of adversarially learned models and develops some understanding of why the model can generalize well on clean data while suffering from robust overfitting.

**Weaknesses:**

1- Some of the assumptions in the study deviate from the standard adversarial training setting. The theoretical framework is limited to a one-layer neural network with a cubic activation function. In addition, the paper assumes that the adversarial training examples are designed for the optimal linear model $ g(x) = \langle w^*  , x\rangle $ at all epochs. As a result, the adversarial perturbation for each training sample is not updated over the course of robust training and will remain the same at every iteration. These assumptions seem different from a standard adversarial training setting where the perturbations are optimized for the neural net parameters at the current iteration. To address this issue, the theoretical framework should be updated so that the adversarial examples are designed for the current iterate $g(x)=\langle w^{(t)} ,x\rangle$ ($w^{(t)}$ rather than optimal $w^*$).

2- The numerical experiments look insufficient and do not consider the theoretical framework studied in the paper. The paper should include some numerical results on the framework studied in the paper (one-layer neural net with cubic activation), and the text should better analyze and explain the experiments on MNIST and CIFAR-10. Also, the presentation of the numerical results could be improved. The plots in Figure 2 are quite small and hard to read. They do not contain the accuracy of trained models. Overall, section 6 is relatively short and has little discussion on the interpretation of numerical results. I suggest the authors remove section 4.3 from the main text and use more space for section 6, the plots should be presented more clearly and the numerical results should be discussed in greater detail in the text.

Also, Figure 1 in the paper copies a figure from (Rice et al, 2020) which is inappropriate for an independent paper. I think the authors should either only cite Figure 1 from (Rice et al, 2020) in the text or reproduce a similar plot on a slightly different classification setting to meet the expected novelty requirements in the presented numerical results.

**Questions:**

1- The theoretical results focus on the cubic activation function. Do the results hold for other polyonmial activation functions?

2- Parameterization 3.1 requires the adversarial training data to be created for the optimal linear model with $w^*$ at every training iteration. Do the theoretical results remain true if the adversarial training data are optimized for the model at the current iteration $g(x)=\langle w^{(t)} , x\rangle$?

---

> ### Author Response · Authors · 2023-11-22
> **Response to Reviewer CpMF:**
>
> We thank the reviewer of the insightful feedback. We address the reviewer’s concerns as below, and we are happy to clarify any follow-up questions/confusions.
>
> **Q1** The theoretical results focus on the cubic activation function. Do the results hold for other polyonmial activation functions?
>
> **A1** Different from some previous work analyzing the behavior of linear classifier in adversarial training, we improve the expressive power of model by using non-linear cubic activation that can characterize the data structure, in order that we can analyze learning process more precisely. Moreover, under our theoretical framework, the results on feature learning process still hold for more general activation function $z^{q}$, where q is arbitrary odd number at least three to balance the positive-labeled data and negative-labeled data.
>
> **Q2** Parameterization 3.1 requires the adversarial training data to be created for the optimal linear model with $w^{*}$ at every training iteration. Do the theoretical results remain true if the adversarial training data are optimized for the model at the current iteration $g(x) = <w^{(t)}, x>$?
>
> **A2** Theoretically, traditional formulation of adversarial training algorithms is a highly non-convex and non-concave min-max optimization, under which, in general, analyzing global convergence is NP-hard problem [1]. To overcome this hardness barrier, we leverage geometry-inspired attack as adversarial training method, under which we propose a non-trivial three-stage analysis technique to decouple the complicated feature learning process as follows.
>
> &emsp;**Phase I**: At the beginning, the signal component of lottery tickets winner $max_{r\in [m]}u_{r}^{(t)}$ increases quadratically (Lemma 4.2). At this point, the model starts to learn partial true feature.
>
> &emsp;**Phase II**: Once the maximum signal component $max_{r\in [m]}u_{r}^{(t)}$ attains the order $\tilde{\Omega}(\alpha^{-1})$, the growth of signal component nearly stop updating since that the increment of signal component is now mostly dominated by the noise component (Lemma 4.3).
>
> &emsp;**Phase III**: After that, by the quadratic increment of noise component (Lemma 4.4), the total noise $V_{i}^{(t)}$ eventually attains the order $\Omega(1)$, which implies the model memorizes the spurious feature (data-wise noise) in final.
>
> &ensp;**On the empirical side**, it is verified that it has a comparable performance with classical adversarial training methods such as FGSM and PGD attacks (e.g., [2][3]). Understand the feature learning process of FGSM and PGD attacks is an existing future direction.
>
> [1] Some NP-complete problems in quadratic and nonlinear programming.
>
> [2] Deepfool: a simple and accurate method to fool deep neural networks.
>
> [3] Geometry-inspired top-k adversarial perturbations.

---

### Meta-Review · Area_Chair_UifP · 2023-12-08

**Metareview:**

The paper studies the overfitting phenomenon in the context of adversarial training and attempts to understand how clean generalization and robust overfitting could happen at the same time. The reviewers find that the assumption of the patch data is too strong which is unrealistic in the real-world applications. and the the choice of the activation function is questionable. More experiments are also needed.

**Justification For Why Not Higher Score:**

N/A.

**Justification For Why Not Lower Score:**

N/A.

---

### Decision · Program_Chairs · 2024-01-16

Reject